# The validity of small-sided games in predicting 11-vs-11 soccer game performance

**Tom L. G. Bergkamp**[1]*, **Ruud J. R. den Hartigh**[2], **Wouter G. P. Frencken**[3,4], **A. Susan M. Niessen**[1], **Rob R. Meijer**[1]

**1** Department of Psychometrics and Statistics, Faculty of Behavioral and Social Sciences, University of Groningen, Groningen, the Netherlands, **2** Department of Developmental Psychology, Faculty of Behavioral and Social Sciences, University of Groningen, Groningen, the Netherlands, **3** Center for Human Movement Sciences, University of Groningen, University Medical Center Groningen, Groningen, the Netherlands, **4** Football Club Groningen, Groningen, the Netherlands

* T.L.G.Bergkamp@rug.nl

**Data Availability Statement:** Our data can be accessed through a Dataverse repository (https://hdl.handle.net/10411/XEVAVU). The data contains sensitive information of our participants. Because

## Abstract

Predicting performance in soccer games has been a major focus within talent identification and development. Past research has mainly used performance levels, such as elite vs. non-elite players, as the performance to predict (i.e. the criterion). Moreover, these studies have mainly focused on isolated performance attributes as predictors of soccer performance levels. However, there has been an increasing interest in finer grained criterion measures of soccer performance, as well as representative assessments at the level of performance predictors. In this study, we first determined the degree to which 7-vs-7 small-sided games can be considered as representative of 11-vs-11 games. Second, we assessed the validity of individual players' small-sided game performance in predicting their 11-vs-11 game performance on a continuous scale. Moreover, we explored the predictive validity for 11-vs-11 game performance of several physiological and motor tests in isolation. Sixty-three elite youth players of a professional soccer academy participated in 11 to 17 small-sided games and six 11-vs-11 soccer games. In-game performance indicators were assessed through notational analysis and combined into an overall offensive and defensive performance measure, based on their relationship with game success. Physiological and motor abilities were assessed using a sprint, endurance, and agility test. Results showed that the small-sided games were faster paced, but representative of 11-vs-11 games, with the exception of aerial duels. Furthermore, individual small-sided game performance yielded moderate predictive validities with 11-vs-11 game performance. In contrast, the physiological and motor tests yielded small to trivial relations with game performance. Altogether, this study provides novel insights into the application of representative soccer assessments and the use of continuous criterion measures of soccer performance.

## Introduction

Professional soccer organizations strive to identify, select, and develop players who have the potential to become elite soccer players. In order to establish evidence-based selection

our sample was relatively small, and the club and team age categories can be derived from the author affiliations and manuscript text, this might lead to indirect identification. In consultation with the Ethical Committee Psychology of the University of Groningen, two restrictions on openly sharing the data were therefore applied. First, data can only be accessed by qualified researchers, that is, researchers affiliated with universities or independent, non-commercial research institutes. Second, researchers must sign a confidentiality agreement, stating that the downloader does not share the data with persons who are not collaborators on the project the data are used for. Requests will be handled by a staff member who is not one of the authors.

**Funding:** This research was partially funded by the Royal Dutch Football Association (Koninklijke Nederlandse Voetbalbond, KNVB, www.knvb.com). Moreover, a commercial organization, Football Club Groningen, facilitated the research and provided support in the form of a salary for one author; WGPF. The KNVB and Football Club Groningen did not have any additional role in the study design, data collection and analysis, decision to publish, or preparation of the manuscript. The specific roles of the authors are articulated in the 'author contributions' section.

**Competing interests:** Football Club Groningen provided funding, in the form of a salary, to one of the authors (WGPF) and facilitated that the research could be conducted at their club. This affiliation does not alter our adherence to PLOS ONE policies on sharing data and materials. Please note that we determined two terms of data access in consultation with the Ethical Committee Psychology of the University Groningen. First, data are only available to qualifying researchers, that is, researchers affiliated with universities or independent, noncommercial research institutes. Second, researchers must sign a confidentiality agreement, stating that the downloader does not share the data with persons who are not collaborators on the project the data are used for. These terms are also specified in our data availability statement.

procedures, talent selection and identification studies often aim to determine the extent to which distinct skills and abilities are related to future performance [1, 2]. This has led to a plethora of studies examining the predictive value of many different kinds of attributes across different performance domains, such as height and weight (i.e., anthropometric attributes), sprint speed, endurance capacity, and agility (i.e., physiological and motor skills), dribbling and passing skills (i.e., technical skills), and motivation and self-regulation (i.e., personality-related or psychological) [3–6]. These attributes are typically assessed in laboratory settings or field tests, and in isolation of in-game soccer constraints [7]. Moreover, the value of these attributes as indicators of 'talent' is assessed by examining how well they discriminate between players with different (future) performance levels (e.g., elite versus non-elite players), or between selected and deselected academy players [8]. As discussed below, the way the predictors and criterion-performance have been defined in previous studies has limitations. Consequently, there has been an increasing interest in finer grained criterion measures of soccer performance, and more ecologically valid assessments at the level of performance predictors [1, 3, 8–11].

## Soccer performance criterion

Using performance levels as the criterion (i.e., the outcome variable and performance to predict) is understandable from a practical standpoint, but has a few disadvantages [8]. First, a disadvantage of this approach is that there are often inconsistencies in the definition of performance levels, which may impede comparisons across studies. For example, definitions of elite athletes have ranged from international to regional level competitors, and strongly depend on the competitiveness of the sport in the athlete's country [12]. Second, since talent research ultimately aims to identify players who have the potential to excel in soccer games [10], it can be argued that the environments of interest are competitive 11-vs-11 games. It follows that the relevant criterion is, ideally, individual performance within these games [8, 10]. However, while coaches or scouts—responsible for grouping players into performance levels—arguably decide what talented in-game performance looks like, the validity of these judgments is not well established, and is often even biased [11, 13, 14]. For instance, judges (e.g., coaches) are easily influenced by factors unrelated to performance, such as the athlete's appearance or reputation [15, 16]. The bias of coaches to select more mature players, or players born earlier in the calendar year, has also been well established in soccer [17]. Finally, and importantly, dichotomizing the criterion into performance levels provides no information on the differences between individuals within the same level on an in-game soccer performance outcome [8, 18]. Therefore, talent identification researchers are facing the question whether they can define in-game soccer performance criteria that are not based on grouping performance levels, and that are able to distinguish between individual players on a continuous scale [8, 10, 19, 20].

There are multiple ways to quantify different aspects of individual in-game soccer performance. Global and local positioning systems may be used to quantify physiological in-game performance characteristics, such as high intensity meters, total distance run, and accelerations [21]. By extracting spatio-temporal information of the players on the pitch, these systems may also be used to assess tactical performance indicators, such as the space created with a pass [22]. A more straightforward technique that does not demand advanced technologies is notational analysis. This technique lends itself particularly well to assess on-ball technical and tactical performance indicators, by manually coding observed events [23, 24]. Recent work suggests that performance indicators derived through this technique, such as passes, duels, and shots, are related to game success (i.e., winning) [25]. This opens promising opportunities for

operationalizing soccer performance at the criterion level, as well as assessing performance at the predictor level.

## Assessments in soccer

The attributes assessed in the talent identification literature resulted in various levels of success in discriminating between performance levels [5, 26]. For example, a recent systematic review evaluated the discriminatory value of different physical and physiological attributes [6]. The authors found median effect sizes across studies of $d = 0.37$ for sprint speed ($< 20$m), $d = 0.41$ for endurance capacity, and $d = 0.42$ for change of direction, which can be considered low [27]. In contrast, repeated sprinting ability and sprint speed ($> 20$m) had effect sizes of $d = 1.21$ and $d = 0.57$, which can be considered as strong and medium, respectively.

Nevertheless, it has recently been argued that assessments that are representative of competitive 11-vs-11 games may result in better performance predictions compared to abilities that are tested in isolation [7, 8, 11, 28–30]. Representative assessment is described as a design that maintains, or 'samples', the personal, environmental, and task constraints of the performance environment of interest [28, 29]. When the criterion is operationalized as performance in 11-vs-11 games, a representative context incorporates environmental constraints in these games, such as the presence of moving opponents and the task to score goals. At the same time, it simulates soccer-specific motor, physiological, technical, tactical, and perceptual-cognitive in-game performance behaviors for the player [8, 11, 31]. Thereby, representative assessments do justice to the idea that the mechanism underlying elite soccer performance is characterized by how the player acts upon, and interacts with environmental constraints [11].

By simulating 11-vs-11 games, a representative assessment also builds on the notion of behavioral consistency. That is, the assumption that the best predictor of future behavior is similar behavior in the past [32, 33]. Predictors that are similar to the criterion in content and context are said to be high in fidelity. Accordingly, research in sports has repeatedly demonstrated that predictive validity increases when the fidelity of the predictor increases [34–36]. Tests that measure attributes that are less similar to the criterion behavior (i.e., 11-vs-11 game performance) may be considered as lower-fidelity attributes [28, 34, 37]. From this point of view, representative assessments would provide higher-fidelity predictors than tests measuring motor, physiological, technical, tactical, and perceptual-cognitive attributes in isolation.

An example of representative assessments in soccer are small-sided games (SSGs) [10, 11, 38]. SSGs are games played with fewer players and on a smaller pitch size compared to 11-vs-11 games. However, the degree of representativeness may be dependent on variations in the specific number of players and pitch size [39]. It is, therefore, important to evaluate the degree to which SSGs are representative of 11-vs-11 game. To the best of our knowledge, one study has been conducted in this direction. Results from Olthof et al. [40] suggest that the tactical demands of SSGs for under-13 year old (U13), U15, U17, and U19 players reflect those of 11-vs-11 games, when teams consist of 6 or 8 players and when a match derived relative pitch area of 320 m$^2$ per player is used.

Interestingly, the few studies that have explored the concurrent or predictive validity of individual SSG performance mainly included smaller SSGs. Fenner et al. [41] and Unnithan et al. [10] showed that 4-vs-4 SSG performance for U10 and U16 players, based on matches won and goals scored, had a strong to moderate relationship with technical skills, as determined by a scouting tool ($r = 0.76$ and $r = 0.39$, respectively). Moreover, Bennett et al. [42] demonstrated that on-ball skill proficiencies, such as dribbles, passes, touches, and shots, discriminated significantly between high and low-level soccer players in 4-vs-4 SSGs. While these studies provide important first clues on how individual SSG performance may be utilized for

performance assessment, an exploration of performance in larger SSGs as predictors of performance in 11-vs-11 games has not been conducted yet. Furthermore, the previous studies correlated overall SSG performance with subjective scout ratings or performance levels [10, 41, 42], whereas more objective in-game indicators may better serve as a criterion measure.

### The current study

The current study expands the previous literature by quantifying in-game soccer performance on a continuous scale. By doing so, we first examined the degree to which performance indicators in large-scaled, 7-vs-7 SSGs can be considered representative of performance indicators in competitive 11-vs-11 games. The concept of representative assessment suggests that predictive validity is driven by using predictors that are highly representative for the criterion. Therefore, the representativeness of SSGs for 11-vs-11 games can be considered a prerequisite for their predictive validity. Second, we explored the value of the SSGs as a high-fidelity predictor, by assessing the validity of individual players' in-game SSG performance in predicting their 11-vs-11 game performance. In addition to our two primary aims, we explored the validity of physiological and motor attributes that are frequently used in the talent literature and by soccer teams in monitoring and predicting performance, namely sprint, agility, and endurance capacity tests [43, 44]. Because these tests may be considered as low-fidelity in relation to individual performance in soccer games, relatively low correlations with the criterion could be expected.

## Materials and methods

### Participants

Elite youth players from the U15, U17, U19, and U23 teams of a professional soccer academy in the Netherlands were recruited to participate in the study. Recruitment started two months before the start of the 2018–2019 competitive soccer season, and was conducted after approval from the youth players, the coaches, the academy's technical director and the club's head of performance. All players belonging to the U15 to U23 teams were eligible to participate in the study, resulting in $n = 87$ who participated in at least one SSG over the course of the season. However, we excluded players who did not play any minutes in the 11-vs-11 games or played in few SSGs (i.e. more than 2 standard deviations below the average number of SSGs played per team; see Table 1), due to injury, dropping out of the academy, or other circumstances. This resulted in a total of $n = 63$ players from the U15 ($n = 17$), U17 ($n = 15$), U19 ($n = 16$), and U23 ($n = 15$) teams who were included in the analyses.

Table 1 presents descriptive information of the included players per team. The players of the different teams had comparable practice schedules. They had four or five technical and tactical practice sessions and one or two physical practice session per week, resulting in 7.5 to 10.5 hours of practice per week. Additionally, the teams played one competitive match each week. The U17 and U19 teams competed at the highest and second highest national level

**Table 1. Descriptives (mean, SD in brackets) for the elite players ($n = 63$) included in the study, classified by age category (i.e., team).**

| Team | n | Age (yrs) | Height (cm) | Weight (kg) | SSGs (number) | Playing time SSG (min) | Playing time 11-v-11 (min) |
|------|-----|-----------|-------------|-------------|---------------|------------------------|----------------------------|
| U15 | 17 | 14.04 (0.40) | 161.29 (5.85) | 47.29 (5.18) | 16.00 (4.51) | 96.00 (27.08) | 127.00 (71.78) |
| U17 | 15 | 15.97 (0.58) | 176.60 (7.57) | 64.01 (7.16) | 11.47 (2.20) | 68.80 (13.22) | 162.80 (91.13) |
| U19 | 16 | 17.45 (0.39) | 181.94 (7.47) | 70.34 (8.83) | 17.75 (4.80) | 106.50 (28.77) | 131.25 (71.21) |
| U23 | 15 | 19.41 (1.05) | 181.29 (5.18) | 74.74 (7.38) | 14.80 (3.97) | 88.80 (23.81) | 153.53 (70.55) |

within their respective youth competition, the U15 team competed at the third highest national level. Players in the U23 team competed at the highest adult amateur level. Thus, participants in this study played at an elite level given their age, and our sample is considered to be representative of the population of elite soccer players in the U15 to U23 age categories. Written informed consent was acquired from the players (and their parents when necessary) prior to the start of the study. The protocol of the study was approved by the Ethical Committee of Psychology, University of Groningen (Research code: 17197-O).

## Procedure and measures

**Predictor: SSGs.** The SSGs for this study were organized approximately once per month, over the course of 8 months, as part of the regular technical and tactical training sessions for each team. The SSGs were scheduled in consultation with the teams' physical trainers. Depending on the physical load scheduled for the teams by the physical trainers, 3 to 6 SSGs per team were organized per training session. Due to uncontrollable circumstances, such as the cancellation of training sessions due to bad weather, the absence of players due to illness or injuries, or players dropping out, players within and across teams could not participate in the exact same number of SSGs. Therefore, players in the U15, U17, U19, and U23 teams played on average in 16, 11, 17, and 14 SSGs, respectively (see Table 1).

The SSGs were played outdoors on the teams' usual practice grounds, with the U23 and U19 teams playing on natural turf and the U17 and U15 teams playing on artificial turf. The pitch size was constrained to 80 m x 56 m, which corresponds to the match-derived relative pitch area of 320 m$^2$ [40]. Each SSG lasted 6 minutes, with 2 minutes of rest in between SSGs, and included standard soccer rules, such as throw-ins, off-side, free kicks, and corner shots. The games were filmed using a Canon Legria HF R68.

Finally, to control for the strength of opposition and the quality of the team, players were reorganized into different teams after each SSG (cf. Fenner et al. [41]). This was done semi-randomly, by accounting for the position (i.e., attack-midfield-defense) of the players in order to avoid teams consisting of mainly one playing position. Thus, players played each game with a different set of teammates.

We used notational analysis to assess performance in the SSGs [24]. A coding scheme detailing offensive and defensive indicators was developed by the first author and the soccer club's head of performance and data analyst. The head of performance and the data analyst each had more than 7 years of experience managing, processing, and analyzing event data (i.e. data on soccer performance indicators, regardless of outcome). The coding scheme contained performance indicators that are positively correlated with game success [25], and were deemed to present an accurate picture of an individual's in-game on-ball performance, namely passes forward, offensive and defensive duels, assists, key passes, shots on target, applying pressure, and pass interceptions (see S1 Table).

Performance indicators in the SSG videos were coded independently by one researcher and two graduate students using Noldus The Observer XT (Noldus Information Technology, Wageningen, the Netherlands). The researcher and graduate students prepared and practiced with coding for a week, in order to make slight adjustments to the definitions of performance indicators and obtain familiarity with the coding scheme. Then, three of the total $k = 82$ SSGs were coded by both the researcher and the students to assess the reliability between the raters. This yielded a Cohen's kappa of 0.77, which indicates acceptable reliability.

**Predictor: Physiological and motor tests.** Physiological and motor testing was conducted approximately two months after the beginning of the season. Players' sprinting ability was measured by a maximal 30-meter linear sprint, with a local position measurement system

tracking the position and time of the players (Inmotio Object Tracking BV, Amsterdam, the Netherlands). Timing gates were placed at the 0, 10, and 30 m mark. Players positioned themselves 0.5 m behind the first timing gate, and were instructed to run as fast as possible. Each player performed 2 sprints. The fastest time was recorded and used for analysis [44].

To assess each athlete's interval endurance capacity, players performed the Interval Shuttle Run Test (ISRT) [45]. During this test, players were required to run back and forth on a 20 m course, with pylons set 3 m before the turning lines. Sound signals on a prerecorded disc indicated the pace at which the players had to reach the 3 m turning lines. The running speed, dictated by the frequency of these signals, was increased by 1 km/hr every 90 s from a starting point of 10km/hr and by 0.5 km/hr every 90 s from 13 km/hr onwards. Each 90 s period was divided into two 45 s periods in which players ran for 30 s and walked for 15 s. Players were instructed to complete as many tracks as possible, and were told to stop when they could not follow the pace or felt unable to complete the run. The maximum number of completed tracks was recorded and used for analysis.

Finally, players' agility was measured using a modified version of the agility T-test [46, 47]. Four cones were arranged in a T shape, with a cone placed 5 m from the starting cone and 5 m on either side of the second cone. Players were instructed to sprint from the starting cone to the second cone, sprint to a side-cone, sprint to the opposite side-cone, sprint back to the second cone, and finally sprint back to the starting cone. This test was conducted twice, with players turning either right or left around the cones, to obtain a right and left agility estimate, respectively. Thus, in this modified version, players had to sprint around, instead of shuffle between the outer cones. Times were recorded using the local position measurement system. An average agility estimate was computed by taking the mean of the left and right estimate, which was used for further analyses.

**Criterion: 11-vs-11 games.**   Criterion data was obtained by analyzing participants' performance in 11-vs-11 games. The 11-vs-11 games were played as part of the team's regular competitions, and were filmed by a staff member of the club. In deciding the number of 11-vs-11 games to analyze, we aimed to match approximately the number of analyzed minutes in the SSGs and 11-vs-11 per team. This would result in analyzing three full 11-vs-11 games per team. However, in order to have sufficient variability in opponent strength, as well as in the performance of the participants, we instead analyzed one half of six different 11-vs-11 games.

Games were selected based on each team's placement in their competition standings: we selected two games against higher placed opponents, two games against lower placed opponents, and two games against opponents with approximately the same placement. For each game we randomly selected either the first or second half. All selected games were played in the last four months of the same season in which the SSGs were played.

Individual soccer performance in the 11-vs-11 games was assessed using the same notational analysis procedure and coding scheme as for the SSGs. Thus, we coded the same performance indicators in the 11-vs-11 games as in the SSGs. The coding process was conducted by the same researcher and graduate students.

## Data preparation

The performance indicators 'dribbles' and 'take-ons' were summed to create an 'offensive duel' indicator; 'tackles' and 'in-fronts' were summed to create an 'defensive duel' indicator (see Table 2). More than half of the players did not have any recorded events on offensive and defensive aerial duels in the SSGs. Therefore, these indicators were excluded from the individual performance analysis.

**Table 2. Definitions and weights for offensive and defensive performance indicators.**

| Indicator | Offense | | | | | |
|---|---|---|---|---|---|---|
| | Pass forward | Dribble | Take on | Chance created | Shot on target (incl. goals) | Offensive aerial duel |
| Definition | A pass attempt in the forward (i.e. opponent's goal) direction. | An attempt by the attacker with the ball to drive by a defender. No dribble is awarded if the attacker dribbles in 'open space' and does not attempt to drive by a defender. | An attempt by the attacker with the ball to maintain ball-control/ possession, and/or create space, when in contest with a defender. | The final pass that leads to the recipient of the ball having a shot on target (i.e. key pass) or scoring a goal (i.e. assist). | A scoring attempt that goes into the net (i.e. a goal) or an attempt that clearly would have gone into the net, but was saved by the goalkeeper or a player who is the last line of defense. | An attempt by the attacker (i.e., the player whose team was in ball possession) to maintain control/ possession of the ball, when in contest with a defender in the air. |
| Merged | - | Offensive duels | | - | - | - |
| Outcome | Successful— unsuccessful | Successful—unsuccessful | | Counted when occurs | Counted when occurs | Successful—unsuccessful |
| Weight[a] | 0.21 | 0.17 | | 0.50 | 1 | - |
| Formula[b] | Offensive performance = Passes forward * 0.21 + Offensive duels * 0.17 + Chances created * 0.50 + 1 * Shot on target | | | | | |
| Indicator | Defense | | | | | |
| | Tackle | Staying in front | | Applying pressure | Interception | Defensive aerial duel |
| Definition | An attempt by the defender to obtain ball control/ possession of an attacking player with the ball | An attempt by the defender to stay in front of an attacking player, in order to prevent a dangerous offensive (e.g., goal scoring) opportunity. | | A situation in which the defender puts pressure on an attacking player with the ball, thereby making the opposing player lose the ball (e.g. through an unsuccessful pass attempt). | A situation in which the defender 'reads' the pass of the opposing player and moves into the line of the intended the pass, thereby intercepting the pass. | An attempt by the defender (i.e., the player whose team was not in possession) to obtain ball control/ possession, when in contest with an attacker in the air. |
| Merged | Defensive duel | | | - | - | - |
| Outcome | Successful—unsuccessful | | | Counted when occurs | Counted when occurs | Successful—unsuccessful |
| Weight[a] | -0.14 | | | -0.11 | -0.06 | - |
| Formula[b] | Defensive performance = (Defensive duels * -0.14 + Interceptions * -0.06 + Applying pressure * -0.11) * -1 | | | | | |

[a] *Weights* indicate the aggregated correlation of the performance indicator with shots on target (offensive) and shots on target conceded (defensive).

[b] *Formula* indicates the computation for the individual overall offensive and defensive performance. Performance indicators in the formula row indicate standardized (z) scores. The defensive score was multiplied by -1 such that a higher score indicates a better defensive performance.

In order to compare performance between players who varied in total minutes played, the indicators that were counted 'when they occurred' (i.e., interceptions, applying pressure, chances created, shots on target) were transformed to a rate statistic, by computing the number of events per bout of six minutes (i.e., the duration of each SSG). To operationalize each player's performance on the indicators that had a successful or unsuccessful outcome (i.e., passes forward, offensive duels, and defensive duels) we applied a rigorous statistical approach. Specifically, we estimated a random intercept multilevel logistic regression model for these indicators in both SSGs and 11-vs-11 games, in which the intercepts were allowed to vary across players. The advantage of this model is that it does not require an equal number of observations for each individual (e.g., simply dividing successful passes by total number of passes may lead to over- or underestimations of a player's performance [48]). In addition to the random intercepts, 'team' was included as a categorical covariate. This model predicts the probability of a successful outcome on the indicator (i.e., the dependent variable, for example, a successful pass) for each player simply by their intercept (i.e., the model's fixed effect intercept plus a random effect for each player) and their team effect. Thus, these 'posterior' estimates can be seen as a measure of each player's performance on the performance indicators (see S2 Table for a summary of the multilevel models).

Finally, we combined the offensive and defensive performance indicators to obtain an overall measure of offensive and defensive in-game performance for each player, respectively. The weights for each indicator were derived from its team-wise correlation with a proxy for in-game offensive and defensive success, namely shots on target and shots on target conceded (i.e., a shot on target by the opposite team, both including goals; cf. Pappalardo et al. [49]). Specifically, we assessed the *team's* performance on the performance indicators in each SSG and 11-vs-11 game, and computed Spearman's rank correlations between the indicators their respective in-game success proxy (see Table 1 and S3 Table). To account for differences in the number of observations and performance levels across age groups, the correlations were aggregated using a random effect meta-analysis. The correlation coefficients for each indicator were in the expected direction, meaning that greater performance on the offensive indicators was positively associated with shots on target, while greater performance on the defensive indicators was negatively associated with shots on target conceded (see Table 1). Therefore, we transformed the performance indicators for the players to z-scores within each team, multiplied their score with the correlation coefficient, and summed the scores [49]. Additionally, we added the individual player's shots on target to the offensive performance measure, giving it a weight of 1. These overall performance measures can be seen as a player's contribution to in-game success.

## Statistical analyses

To evaluate the extent to which SSGs are representative for 11-vs-11 games in terms of the assessed performance indicators (i.e., aim 1), we first computed the mean number of times an event occurred per 6 minutes of playing time, for each performance indicator, in each game format. Second, we conducted a chi-square goodness of fit test to compare the total number of observed events per performance indicator in the SSGs (i.e. the empirical distribution) against the relative frequency of the observed events on the performance indicators in the 11-vs-11 games (i.e. treating this as the theoretical distribution). We checked the observed and expected events, as well as the Pearson standardized residuals to evaluate which performance indicators differed most in incidence in the SSGs and 11-vs-11 games. Given that effect sizes for chi-square tests are often difficult to interpret [27], we computed a Spearman's rank correlation ($r_s$) between the total number of observed events in both game formats to assess the degree of association between the distributions.

To assess the predictive validity of SSG performance (i.e., aim 2), we computed Spearman's rank correlations between the performance indicators in the SSGs and 11-vs-11 games. Moreover, to assess the predictive validity of physiological and motor performance, we computed Spearman's rank correlations between the physiological and motor tests and overall offensive and defensive performance in the 11-vs-11 games. Players with partially missing data (i.e., on either the ISRT, sprint, or agility tests) were still included in analyses for which they had sufficient data. Four players did not have enough offensive duel events and 2 players did not have defensive duel events in the 11-vs-11 games. In addition, 6 players could not participate in the sprint- and agility tests due to illness or injury, including 1 that could also not participate in the ISRT. One player had missing data on both the sprint test and offensive duels. This yielded sample sizes of $55 < n < 63$ for the different analyses.

To account for possible differences between players across teams, correlations were first computed within each team. Then, in order to draw inferences on the overall strength of the predictor-criterion relationships across our sample ($55 < n < 63$), we combined the coefficients from the different teams using a random effect meta-analysis. The random effect meta-analysis accounts for the heterogeneity across coefficients, as well the sample size per team,

resulting in a weighted average correlation coefficient [50]. We refer to the weighted average coefficients as the aggregated correlation coefficient.

We computed Spearman's rank correlations instead of Pearson correlations, because we are interested in the association between the rankings on the predictors and criterion, and want to account for any potential outliers. The correlations' magnitudes were interpreted according to the thresholds suggested by Cohen [27], with $r_s$ = 0–0.1 indicating a trivial, $r_s$ = 0.1–0.3 indicating a small, $r_s$ = 0.3–0.5 a moderate, and $r_s$ > 0.5 a large relationship. Finally, while we report $p$-values, we aim to avoid dichotomizing results as 'significant' or not, and focus on the point estimates and confidence intervals [51, 52].

## Results

### Representativeness of SSGs

Fig 1 presents the mean number of events per 6 minutes for each performance indicator, per SSG and 11-vs-11 game (see S4 Table for a table with this information). With the exception of aerial duels and pass interceptions, there were more events per 6 minutes for every performance indicator in an average SSG, compared to an average 11-vs-11 game.

Table 3 presents results from the chi-square goodness of fit test. The chi-square goodness of fit test indicated that the total number of observed events per indicator in the SSGs was not consistent with the distribution of events in the 11-vs-11 games, $\chi^2$ (10, N = 6060) = 923.79, $p$ < 0.01. By examining the expected number of events and the standardized residuals in Table 3, it can be seen that this finding is mainly driven by both aerial duels, the shots on

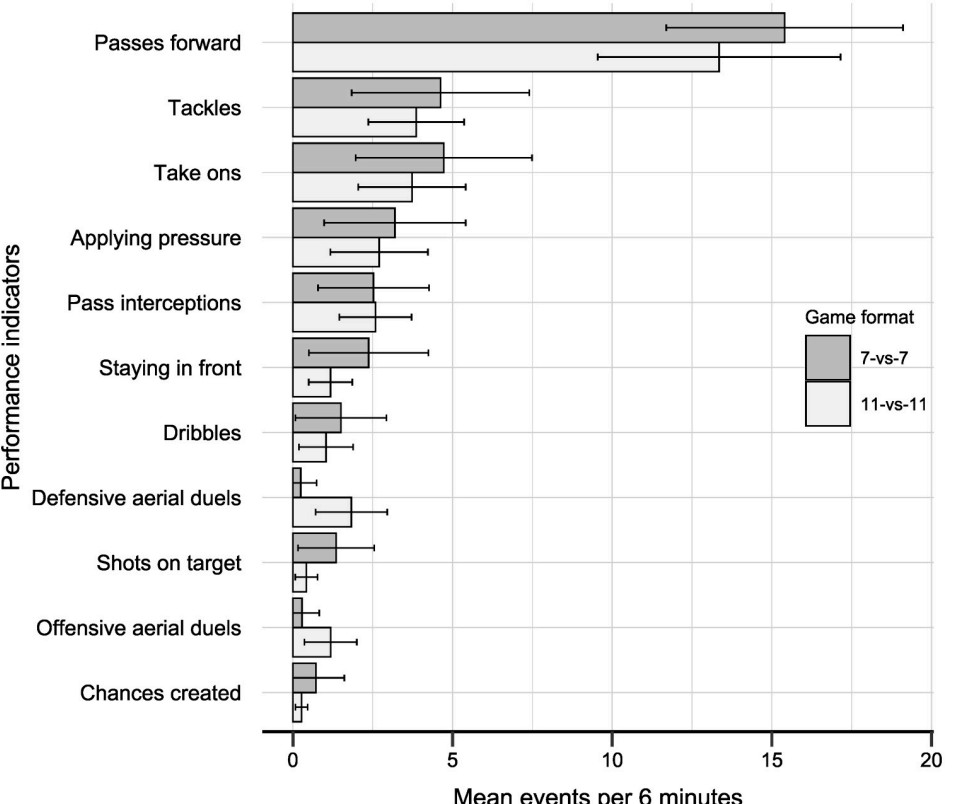

**Fig 1. Mean events per 6 minutes for the performance indicators in 7-vs-7 SSGs and 11-vs-11 games.**

**Table 3. Results from the chi-square goodness of fit test.**

$\chi^2$ (10, $N = 6060$) = 923.79, $p < 0.01$

| Performance indicator | Observed events 11-vs-11[a] | Prop.11-vs-11 | Observed events SSG[a] | Prop. SSG | Expected events SSG | St. residuals |
|---|---|---|---|---|---|---|
| Passes forward | 2167 | 0.416 | 2526 | 0.417 | 2519.09 | 0.18 |
| Tackles | 619 | 0.119 | 758 | 0.125 | 719.57 | 1.53 |
| Take-ons | 601 | 0.115 | 775 | 0.128 | 698.65 | 3.07 |
| Applying pressure | 439 | 0.084 | 524 | 0.086 | 510.33 | 0.63 |
| Pass interceptions | 418 | 0.08 | 414 | 0.068 | 485.92 | -3.40 |
| Defensive aerial duel | 303 | 0.058 | 40 | 0.007 | 352.23 | -17.14 |
| Staying in front | 195 | 0.037 | 389 | 0.064 | 226.68 | 10.99 |
| Offensive aerial duel | 195 | 0.037 | 47 | 0.008 | 226.68 | -12.16 |
| Dribbles | 165 | 0.032 | 247 | 0.041 | 191.81 | 4.05 |
| Shots on target | 68 | 0.013 | 222 | 0.037 | 79.05 | 16.18 |
| Chances created | 43 | 0.008 | 118 | 0.019 | 49.99 | 9.66 |

Prop = proportion; st. = standardized.

[a]used to assess the correlation between the distribution of events in both game formats.

target, chances created, and staying in front. Specifically, there were substantially fewer aerial duels in the SSGs than in the 11-vs-11 games, whereas shots on target, chances created and staying in front were observed more often in the SSGs (see also Fig 1). However, while there were differences on these performance indicators between the observed and expected events, we found that the overall association between the distributions was strong ($r_s$ = 0.78, 95% CI = 0.35–0.94). The overall high degree of representativeness of the SSGs is also supported by the finding that the removal of aerial duels reduces the chi-square value by approximately a half ($\chi^2$ (8, $N = 5973$) = 422.52, $p < 0.01$), and increases the correlation to $r_s$ = 0.98, (95%, CI = 0.92–1). Together, these results suggest that, with the exception of aerial duels, the distribution of events is similar in the SSGs compared to the 11-vs-11 games. However, the SSGs yield more opportunities for events on the performance indicators, particularly in terms of shots on target and chances created.

## Individual SSG performance

Table 4 displays the aggregated Spearman's correlations between the players' performance on the different indicators in the SSGs and the 11-vs-11 games (see S5 Table for correlations per team). With respect to the aggregated coefficients, individual performance in the SSGs and 11-vs-11 games was moderately-to-largely correlated for 6 of the 9 performance indicators. The largest relationship was found for performance on pass interceptions ($r_s$ = 0.53, 95% CI = 0.25–0.73). Individual forward passing performance ($r_s$ = 0.38, 95% CI = 0.11–0.59), offensive duel performance ($r_s$ = 0.35, 95% CI = 0.08–0.58), shots on target ($r_s$ = 0.38, 95% CI = 0.05–0.63), successfully applying pressure ($r_s$ = 0.40, 95% CI = 0.13–0.61), and overall offensive performance ($r_s$ = 0.46, 95% CI = 0.20–0.65) in the SSGs and 11-vs-11 games were moderately correlated. A small correlation was found for overall defensive performance ($r_s$ = 0.28, 95% CI = 0–0.52), while trivial correlations were found for defensive duel performance ($r_s$ = 0.02, 95% CI = -0.26–0.30) and chances created ($r_s$ < 0.01, 95% CI = -0.27–0.26). Moreover, the confidence intervals for every indicator were relatively wide, ranging from a positive small to positive large association for the indicators with a moderate-to-large point estimate. In sum, these results suggest that the predictive validity of individual SSG performance is moderate-to-large but that there is variability across performance indicators.

**Table 4. Aggregated Spearman's correlations between the performance indicators in the SSGs and 11-vs-11 games.**

| Performance indicator | $r_s$ (95% CI) | $p$ | $n$ |
|---|---|---|---|
| Forward passing | 0.38 (0.11–0.59) | 0.007 | 63 |
| Chances created | < 0.01 (-0.27–0.26) | 0.98 | 63 |
| Shots on target | 0.38 (0.05–0.63) | 0.03 | 63 |
| Pass interceptions | 0.53 (0.25–0.73) | < 0.001 | 63 |
| Applying pressure | 0.40 (0.13–0.61) | 0.005 | 63 |
| Offensive duels | 0.35 (0.08–0.58) | 0.01 | 59 |
| Overall offensive performance | 0.46 (0.20–0.65) | < 0.001 | 59 |
| Defensive duels | 0.02 (-0.26–0.30) | 0.88 | 61 |
| Overall defensive performance | 0.28 (0–0.52) | 0.05 | 61 |

$r_s$ = aggregated spearman correlation coefficient; CI = Confidence Interval.

## Physiological and motor performance

Table 5 presents Spearman's correlations between the players' performance on the physiological and motor tests and the overall offensive performance (left), and the overall defensive performance (right) in the 11-vs-11 games (see S6 Table for correlations per team). The aggregated coefficients were negative small or trivial for 10 m sprint and 11-vs-11 performance ($r_s$ = -0.19, 95% CI = -0.47–0.12; $r_s$ = 0.05, 95% CI = -0.24–0.34), 30 m sprint and 11-vs-11 performance ($r_s$ = -0.20, 95% CI = -0.54–0.20; $r_s$ = 0.02, 95% CI = -0.26–0.31), and agility and offensive performance ($r_s$ = -0.11, 95% CI = -0.46–0.29). A small positive aggregated correlation was found for offensive performance and ISRT ($r_s$ = 0.15, 95% CI = -0.22–0.48). Moreover, a small negative aggregated correlation was found between ISRT and defensive performance ($r_s$ = -0.12, 95% CI = -0.38–0.17), and a small positive correlation for defensive performance and agility ($r_s$ = 0.11, 95% CI = -0.18–0.39). Additionally, the confidence intervals were wide, and ranged from a (small-to-large) negative to (small-to-moderate) positive association for all physiological and motor tests. In sum, the point estimates suggest that the predictive validity of physiological and motor test performance varies between small and negative to small and positive, with respect to our operationalization of overall offensive and defensive performance in the 11-vs-11 games.

## Discussion

In the current study we aimed to take novel steps in quantifying in-game soccer performance, and in assessing the representativeness of SSG performance for 11-vs-11 game performance.

**Table 5. Aggregated Spearman's correlations between physiological and motor tests and overall offensive (left) and defensive performance (right) in 11-vs-11 games.**

| Physiological and motor performance | Overall offensive performance (11-vs-11) | | | Overall defensive performance (11-vs-11) | | |
|---|---|---|---|---|---|---|
| | $r_s$ (95% CI) | $p$ | $n$ | $r_s$ (95% CI) | $p$ | $n$ |
| 10 m sprint | -0.19 (-0.47–0.12) | 0.23 | 55 | 0.05 (-0.24–0.34) | 0.72 | 56 |
| 30 m sprint | -0.20 (-0.54–0.20) | 0.32 | 55 | 0.02 (-0.26–0.31) | 0.87 | 56 |
| ISRT | 0.15 (-0.22–0.48) | 0.43 | 58 | -0.12 (-0.38–0.17) | 0.42 | 60 |
| Agility | -0.11 (-0.46–0.29) | 0.62 | 55 | 0.11 (-0.18–0.39) | 0.45 | 56 |

$r_s$ = aggregated spearman correlation coefficient; CI = Confidence Interval.

Note: a lower time on the sprinting and agility tests indicates a better performance, hence a negative correlation indicates that faster sprinting and agility is related to better overall performance in 11-vs-11.

First, we examined whether 7-vs-7 SSGs provided a representative assessment context for 11-vs-11 games, in terms of various performance indicators. Second, we determined the predictive validity of individual soccer SSG performance with respect to performance in 11-vs-11 games. Moreover, we explored the predictive validity of physiological and motor tests for performance in 11-vs-11 games.

We found strong associations between the distribution of observed events across the performance indicators in both game formats. Additionally, we found that, on average, more events per 6 minutes occur in the SSGs than in the 11-vs-11 games. This was the case for almost all performance indicators, the main exceptions being aerial duels, which occurred considerably more often in the 11-vs-11 games. Together, these results suggest that the SSGs are representative for 11-vs-11 games in terms of assessed indicators, but that they are generally faster paced than 11-vs-11 games. While the relative pitch area was constrained to match those of official games [40], the smaller absolute pitch size and lower number of players may still lead to a faster offensive play, as shown by the increase in shots, chances created, and staying in front of a player on the defensive end. Likewise, an explanation for the exception of aerial duels is that the smaller pitch size changes the environmental constraints of the soccer game. This may alter the affordances, for instance of aerial goal-kick possibilities, which typically result in aerial duels [53, 54]. Although unanticipated, these results can be interesting and relevant to talent identification and development in soccer. Given that high-paced handling is crucial for modern day professional soccer [55], the large scaled 7-vs-7 SSGs may provide ample opportunities as a practice context. It is also plausible that such patterns are reinforced when pitch or team sizes are reduced even further. Therefore, it would be interesting to assess the extent to which small scaled 4-vs-4 SSG, as used in other studies [41, 42], can be considered representative of 11-vs-11 games.

When looking at the predictive validity of SSG performance, performance on pass interceptions, forward passes, applying pressure, shots on target, offensive duels and overall offensive performance were positively and moderately correlated, meaning that individual performance on these indicators in the SSGs was related to performance in the 11-vs-11 games. In contrast, trivial and small correlations were found for performance on chances created, overall defensive performance, and defensive duels. These results suggest that 7-vs-7 SSGs are particularly useful for assessing and predicting offensive 11-vs-11 performance. The small correlation for overall defensive performance seems a logical result of defensive duels: This indicator received the largest weight in creating the defensive performance indicator, but defensive duels in the SSGs and 11-vs-11 games were not correlated.

More generally, the variability in correlations and relatively large confidence intervals across indicators is likely due to the natural variation around in-game technical and tactical performance [56]. While players across age categories played in multiple SSGs and 11-vs-11 games, the sample size in terms of both minutes played and number of players was still relatively small. This could have made it difficult to obtain stable validity estimates for the performance indicators, particularly for chances created, defensive duels, and defensive performance. Still, the moderate predictive validities based on a relatively small sample size are encouraging of using 7-vs-7 SSGs as representative contexts for predicting performance in 11-vs-11 games.

These findings are in accordance with our hypothesis that a predictor that mimics the criterion behavior in content and context enhances predictive validity (i.e., behavioral consistency). This is reinforced by the finding that the physiological and motor tests yielded trivial-to-small correlations with offensive and defensive performance, as assessed through the indicators. These results, therefore, make intuitive and theoretical sense; they suggest that a predictor based on a representative assessment may be more suitable for making predictions than results

of isolated physiological and motor tests, at least when soccer performance is defined in terms of the assessed performance indicators. In sports, these findings correspond to Lyons et al. [36], who studied the predictive validity of physiological and motor performance and collegiate performance on in-game American football performance. The authors found that collegiate performance was a more valid, and more consistent predictor of American Football performance than physiological tests. Furthermore, the trivial correlations for physiological and motor performance are in accordance with Wilson et al. [18], who showed that athletic ability had a very weak association with performance in 11-vs-11 games, as determined by similar performance indicators.

Although the predictive validity of the physiological and motor tests was small in our study, these results do not mean that physiological and motor performance is unimportant for elite soccer performance *in general*. For example, range restriction in the physiological and motor variables likely attenuated their relationship with 11-vs-11 performance. This means that physiological and motor performance is most likely related to soccer performance in the general population of all youth players. However, there is not enough variance in physiological and motor performance among the elite soccer players to meaningfully differentiate between them, as it is likely that the elite players have, explicitly or implicitly, been preselected on these variables [8]. Thus, stronger relationships may have been found if the physiological and motor variables were studied in a more heterogeneous group of players. Note, however, that this same argument holds for the predictive validity of SSG performance.

## Strengths & limitations

In this study, we developed a finer-grained measure of soccer performance. At the same time, our operationalization of soccer performance cannot be considered a 'complete' measure of in-game performance [57, 58]. We measured in-game performance using performance indicators that could be coded based on recordings of games. For instance, we were not able to reliably define off-the-ball movements for each player at each moment [39], or include physiological measures such as high-intensity sprints on the field, or total distance ran. Integrating such (physiological) measures into our on-ball 11-vs-11 performance metrics could have increased the predictive validities of the physiological and motor tests [59]. In addition, note that although off-ball performance actions, such as positioning, deciding, and running actions were not explicitly assessed, they are often intertwined with other indicators we assessed (e.g., forward passes). Furthermore, and more importantly, we focused on on-ball performance, because this has been shown to predict game success in soccer [25]. Our study further supports these findings; we also found positive and negative correlations between the offensive and defensive performance indicators, and shots on target and shots on target conceded, respectively. In contrast, evidence for the relationship between physiological in-game performance indicators and game success has been mixed [60–62].

Other limitations pertain to the notational analysis method used to assess soccer performance. This is a relatively intensive method to assess performance and its reliability depends on a common interpretation of indicators by each coder. Although the reliability was acceptable in our study, it is almost unavoidable that particular definitions of indicators (e.g., 'applying pressure') leave room for interpretation. Additionally, using the same observers to code both the predictor and criterion data could have positively affected the correlations between the indicators. Integrating physiological or tactical information derived through local or global positioning systems into the predictor or criterion may offer more reliable information. This could improve soccer performance assessments, and future research should consider if this is feasible. Furthermore, performance in the SSGs and 11-vs-11 was assessed in a single season,

which could have increased the correlations between performance in both game formats. Finally, while SSG and 11-vs-11 performance was moderately correlated overall, we did not account for positional differences. Thus, more research is needed assessing the extent to which SSG performance transfers to position-specific roles in 11-vs-11 games.

## Conclusion

This study provides encouraging first results on the usefulness of SSG performance in predicting 11-vs-11 game performance. We demonstrated that SSGs are faster paced, but representative of 11-vs-11 soccer games in terms of the distribution of performance indicators. Moreover, we found that the performance indicators are correlated with game success. Based on these correlations, we used a novel approach to quantify overall offensive and defensive in-game performance, and showed that individual SSG performance was moderately predictive of 11-vs-11 performance. Finally, in line with the notion of behavioral consistency, we found that SSG performance yielded higher predictive validities than physiological and motor tests that are often used in soccer science and practice.

The current study provides a novel step in operationalizing the criterion as in-game performance, in relation to predicting performance based on a representative assessment. However, since the predictive validities in SSGs can still not be considered as large based on our result, we would not (yet) recommend solely using scores on SSGs for talent identification and selection purposes. We encourage researchers to further examine the validity of SSGs. More importantly, future researcher should give further emphasis to quantifying in-game soccer performance at the criterion and predictor level, thereby incorporating physiological and tactical (off-the-ball) parameters. We expect that the rapid technological advancements in soccer analytics can be fruitfully used in future research on talent selection.

## Supporting information

**S1 Table. Detailed coding scheme and event definitions of performance indicators.**
(DOCX)

**S2 Table. Multilevel logistic regression analyses for the performance indicators with a successful—Unsuccessful outcome in 7-vs-7 and 11-vs-11 games.** Coeff. = Estimated Regression Coefficient; SD = Standard Deviation; SE = Estimated Standard Error; The reference group for the factor 'Team' is the Under 15 (U15) age category.
(DOCX)

**S3 Table. Spearman's correlations (95% CI in brackets) between the offensive performance indicators and shots on target (top), and defensive performance indicators and shots on target conceded (bottom), per age category and game format.**
(DOCX)

**S4 Table. Mean (and SD) events per 6 minutes on the performance indicators per SSG and 11-vs-11 game, across all age categories (top) and per age category (bottom).**
(DOCX)

**S5 Table. Spearman's correlations (95% CI in brackets) between the performance indicators in the SSGs and 11-vs-11 games, per age category.**
(DOCX)

**S6 Table. Spearman's correlations (95% CI in brackets) between physiological and motor tests and overall offensive (top) and defensive performance (bottom) in 11-vs-11 games,**

**per age category (i.e. team).**
(DOCX)

## Acknowledgments

The authors would like to thank Marieke Timmerman for her helpful suggestions regarding the data analysis, as well as Lilli Schrijber and Sem Otten for assisting in the notational analysis.

## Author Contributions

**Conceptualization:** Tom L. G. Bergkamp, Ruud J. R. den Hartigh, Wouter G. P. Frencken, A. Susan M. Niessen, Rob R. Meijer.

**Data curation:** Tom L. G. Bergkamp.

**Formal analysis:** Tom L. G. Bergkamp.

**Funding acquisition:** Ruud J. R. den Hartigh, Wouter G. P. Frencken, Rob R. Meijer.

**Investigation:** Tom L. G. Bergkamp, Wouter G. P. Frencken.

**Methodology:** Tom L. G. Bergkamp, Ruud J. R. den Hartigh, Wouter G. P. Frencken, A. Susan M. Niessen, Rob R. Meijer.

**Resources:** Wouter G. P. Frencken.

**Supervision:** Ruud J. R. den Hartigh, Wouter G. P. Frencken, A. Susan M. Niessen, Rob R. Meijer.

**Visualization:** Tom L. G. Bergkamp, Ruud J. R. den Hartigh, Wouter G. P. Frencken, A. Susan M. Niessen, Rob R. Meijer.

**Writing – original draft:** Tom L. G. Bergkamp.

**Writing – review & editing:** Tom L. G. Bergkamp, Ruud J. R. den Hartigh, Wouter G. P. Frencken, A. Susan M. Niessen, Rob R. Meijer.

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
