## [Decision Letter · Decision Letter 0]

9 Apr 2020

PONE-D-20-04681

The validity of small-sided games and physical ability tests in predicting soccer performance

PLOS ONE

Dear Mr. Bergkamp,

Thank you for submitting your manuscript to PLOS ONE. After careful consideration, we feel that it has merit but does not fully meet PLOS ONE’s publication criteria as it currently stands. Therefore, we invite you to submit a revised version of the manuscript that addresses the points raised during the review process.

Please respond to the reviewers comments including a point by point rebuttal.

We would appreciate receiving your revised manuscript by May 24 2020 11:59PM. To enhance the reproducibility of your results, we recommend that if applicable you deposit your laboratory protocols in protocols.io, where a protocol can be assigned its own identifier (DOI) such that it can be cited independently in the future. For instructions see: http://journals.plos.org/plosone/s/submission-guidelines#loc-laboratory-protocols

We look forward to receiving your revised manuscript.

Kind regards,

Caroline Sunderland

Academic Editor

PLOS ONE

Journal Requirements:

1. In your Methods section, please provide additional information about the participant recruitment method and the demographic details of your participants. Please ensure you have provided sufficient details to replicate the analyses such as: a) the recruitment date range (month and year), b) a description of any inclusion/exclusion criteria that were applied to participant recruitment, c) a table of relevant demographic details, d) a statement as to whether your sample can be considered representative of a larger population, e) a description of how participants were recruited, and f) descriptions of where participants were recruited and where the research took place.Moreover, please ensure that the statistical analysis performed has been described in adequate details.

2. Thank you for including your competing interests statement; "The authors have declared that no competing interests exist."

We note that one or more of the authors are employed by a commercial company: Football Club Groningen

Reviewers' comments:

Reviewer's Responses to Questions

**Comments to the Author**

1. Is the manuscript technically sound, and do the data support the conclusions?

Reviewer #1: Yes

Reviewer #2: Yes

2. Has the statistical analysis been performed appropriately and rigorously? 

Reviewer #1: Yes

Reviewer #2: No

3. Have the authors made all data underlying the findings in their manuscript fully available?

Reviewer #1: No

Reviewer #2: Yes

4. Is the manuscript presented in an intelligible fashion and written in standard English?

Reviewer #1: Yes

Reviewer #2: Yes

5. Review Comments to the Author

Reviewer #1: Reviewer Comments:

The validity of small-sided games and physical ability tests in predicting soccer Performance: PONE-D-20-04681

GENERAL COMMENTS:

Thank you for your contribution to PLoS One. Overall, the paper which presents a study analysing the validity of performance characteristics of small-sided games for soccer performance within young soccer players falls within the scope of the journal and should be of interest to the readership. The study’s purpose was to validate different performance indicators assessed during small-sided games (in particular technical/tactical indicators) or physical/motor tests (e.g., linear sprint performance, endurance) to predict technical/tactical performance characteristics in 11 vs 11 “real” matches.

Strengths of the study mainly relate to the idea of using continuous soccer performance outcomes of competition matches to measure success in soccer, the attempt to develop a performance score based on various soccer outcomes (even though they include at least in parts subjective observations) as well as the idea to consider game-specific measures(SSG) to predict soccer performance. However, in its current form the manuscript also presents a range of concerns in regard to its content (aims of the study, utilized predictors/criterion) and presentation (particularly structure and description of methodology).

With regards to the content of the paper, some aspects require improvements.

First, please consider (re-)defining the aims of your study more precisely and reflect on your title. In my point of view, purpose 1 (representativity of SSG for 11 vs 11) and purpose 2 (validity of SSG performance in predicting regular game performance) are quite similar. The real distinction between those two is not precisely explained. Consider combining these ideas as they are rather related.

Much more important are my concerns with regard to purpose 3 (validity of physical tests for in-game soccer performance). On the one hand I see an issue with your terminology: as you use sprint, endurance as well as agility tests here, you cannot use “physical tests” for your selection of measures (physiological parts as well as technical skills and speed abilities are included). Further, please provide a rational for your choice. Why did you use these three tests (and not others)? However, I am even more concerned about the general idea/procedure of how you deal with purpose 3. Physiological as well as speed related test performances are assessed and correlated with mainly specific technical on-ball performance indicators (also given as a limitation of your study in l.352 pp). In my point of view, this explains the quite low correlations between your tests and the in-game performance indicators. Therefore, this should not lead to the conclusion that your used tests are not valid to predict soccer performance. In my point of, you should either use other parameters (physiological, physical) on the “criterion level” or (if there is no possibility to assess such data) think about revising/removing this purpose. In every case you should discuss more precisely that there is also validity given to individual physical (and others) tests to predict soccer performance in the introduction and/or discussion. There is various current literature dealing with this (e.g. Emmonds et al. (2016), Höner et al. (2017), Johnston et al. (2018), …).

At the same time, the even higher correlations between SSG and 11 vs 11 performance (almost at the same time point) can be explained: same measures used for SSG and 11-vs 11 protocol which is ok for the analyses of representativeness, but not for a comparison between validity of SSG performance and physical tests in order to predict in-game (technical) soccer performance. Please double-check.

Further, there are a few issues with the description of the methodological process (see also specific comments and general comment with regard to purpose 3). Your materials and methods section has some inconsistencies with regard to structure. Although I like the idea of using various subheadings to generate a structured section, the authors should reflect this choice. As you describe in your “procedures” the way you proceeded with the SSG, physical tests and 11-vs-11 games, I was quite surprised that there was a Measures section in the following where the notational analysis and the score development war described. In my point of view, this information belongs together and should not be separated (perhaps in one “measures”-section with subheadings divided by dependent and criterion variables?).

By the way: you sometimes use 11-vs 11 games, sometimes in-game performance: please be consistent, and more important: define, in the introduction, what you mean by in-game performance.

Furthermore, there is a statistical analysis section missing. As you partly describe which statistical measures were used in the results (and in the notation analysis) section, it is hard to understand for the readership what you exactly did. Please provide a separate section where you describe your analyses in more detail (especially provide rational why you use aggregated rank correlations, see also specific comments on Figure as well as purpose 1.)

SPECIFIC COMMENTS

Title

• Depending on your decision regarding general comments: think about removing the physical tests from the title

Abstract

• L. 25: delete redundancy (research)

• L. 27: representative for? Please incorporate.

• L.29ff: revise, combine purposes (see general comment)

• L.33: “in order to”?

• L.41: “tests”

Introduction

• L.49: Please elaborate and add some information about studies dealing with this.

• L.52-56: Please move this part to “The current study” and combine it with the information given there.

• L.63: please define regular game performance (see also general comment)

• L.76-79: provide reference/evidence

• In my point of view, the prognostic value could not be described as “relatively weak” in this case, as there are medium effect sizes (and various studies I am aware of which state that there is an evidence!). Please revise and add more information (here or in the discussion).

• L.92: ability? Or skills?

• L.92: “very strong” is not the right terminology for a correlation of r = 0.39. Please revise.

• L.94: ad “significantly”?

• L.100ff: provide reference

• L.106-114: please revise and combine with L.52-56 (please, avoid redundancies)

Methods

• L.119: Add „N = 63“.

• L.122: Did you control for the different performance levels of the teams? Please provide information and/or discuss.

• L.132 ff. pleas revise as this part remained unclear.

• L.148: “two months after the beginning of the season”?

• L.149: “linear” instead of straight line?

• L.155: delete “the” and use “each athlete” to avoid redundancy.

• L.165: provide rational for the use of this test to measure agility (as far as I know, this test is not specific to soccer) (see also general comment)

• L.175: delete “the”

• L.176: “that were filmed”? Please revise.

• L.180ff: provide a rational for the use of only one half of the match

• L.184: think about combining with procedures (see also general comment)

• L.189: please provide information about the validity of the utilized coding scheme (level of experts, procedure, …)

• L.192: Can you provide some examples for events in SSGs here?

• L.196: Why only 3 of the 6 matches were rated by 3(?) persons and what happened to the other 3? Provide information.

• Table 1: Definition Offensive aerial duel: add “ball” to possession.

• Table 1: a definition of “shots on target conceded” is missing here. Further, why did you not include this indicator with weight 1 in the Defense-score as you did for offense? Please, explain.

• L.225/226: provide rational for the assessed indicators (see also notational analysis and general comment)

• L.233: did you compute the correlations team-wise as well? Or in general (what I would guess). Did you check whether there are big differences with regard to the correlations?

• L: 238-240: this information should be part of the method section. By the way, why not remove the players who were excluded from analysis from the study sample?

Results:

• Present statistical analysis in the method section separately (see general comment)

• L.244: I struggle if Spearmans rank correlation is the right measure here. I would have expected t-Tests/non parametric adequate here to provide information about differences within the frequencies of events (for single indicators and scores). In doing so, you get information about each individual indicator as well as your score, no?

• L.256: Not clear what this means. Please provide more information. Why did you aggregate team-wise? Is there no influence of age class?

• Figure 1: it would be beneficial to add some inferential statistics here as well (e.g.: results from test of mean/median differences?)

• Figure 1: provide more information about the different categories: are the shots not part of chances created? If so, I wonder why the frequencies of shots in 7-7 is higher than for chances created in 7-7?

• L.265: “59 <= N <= 63”?

• L.266: Please provide more information about the aggregation of coefficients (see also general comment)

• L266-271: move this part to the statistical analyses section.

• L.272: “aggregated”?

• L.273ff: remove redundancy.

• L.285-286: I think this result should be discussed in detail within the discussion section (maybe also by giving perspectives for future research as well as dealing with the limits of the used statistical procedures (correlations).

• Table 2: provide n in every row

• Table 2: would it not be beneficial to present correlations coefficients for each age class as well?

• L.288 pp: see general comment

• L.293/294: “left” and “right” instead of top/bottom.

• L.295: “effectively zero”? check wording.

• L.295pp provide p-values/CIs/information about significance here.

• Table 3: “p” instead of “p-value”?; provide n in every row.

Discussion

• L.304: a discussion about the right measure of “regular game performance” is needed.

• L.306pp: please note, that this only holds true for the indicators you chose (technical, on-ball) during real matches. Please consider and add information

• L.306pp: please also reflect on the fact, that your correlations could also be “that high” because you assessed SSG an 11-11 with the same measures within a quite short time window.

• L.326-329: Is there further evidence? A more nuanced discussion would add to the content here.

• L.330-L337: Although I see and like the authors purpose of this paragraph, I think that it should be extended (as it represents a main issue and at the same time limitation of the studs, see general comment).

• L.352 pp: see general comment

• L.360: please revise according to your decision about your aims and also give some perspectives for future research.

Literature:

Emmonds, S., Till, K., Jones, B., Mellis, M., & Pears, M. (2016). Anthropometric, speed and endurance characteristics of English academy soccer players: Do they influence obtaining a professional contract at 18 years of age? International Journal Of Sports Science & Coaching, 11(2), 212-218

Höner, O., Leyhr, D., & Kelava, A. (2017). The influence of speed abilities and technical skills in early adolescence on adult success in soccer: A long-term prospective analysis using ANOVA and SEM approaches. PLOS ONE, 12(8), e0182211. doi:10.1371/journal.pone.0182211

Johnston, K., Wattie, N., Schorer, J., & Baker, J. (2018). Talent Identification in Sport: A Systematic Review. Sports Medicine, 48(1), 97-109. doi:10.1007/s40279-017-0803-2

Reviewer #2: Reviewers’ comments to the manuscript:

“The validity of small-sided games and physical ability tests in predicting soccer” (PLOS ONE)

Decision: Major revision

The current manuscript is well-written and covers a topic which should be of interest to other researchers within the field of talent identification and development. The presented study investigates the predictive validity and impact of small-sided games (SSG) on soccer performance. Furthermore, the approach of examining the predictive validity of small-sided game performance is novel and addresses a range of previously identified methodological issues. However, at times a lack of clarity with respect to the presented content makes the manuscript hard to follow and therefore needs to be addressed before potentially considering this paper for publication. Thus, in the following, you will find some general and specific recommendations which are thought to help improving this paper:

GENERAL COMMENTS

• In general, a more consistent use of terminology throughout the entire manuscript is necessary. For instance, better use the same term for your criterion (i.e. in-game performance vs. regular soccer game performance vs. performance in 11 vs. 11 games).

• With respect to your proposed criticisms it is not clearly stated what added value the first one has over the second. My suggestion would be to combine the two and thus only state 2 objectives. In addition, while your methods of validating the SSG is well thought through and practical, your way of examining the predictive validity of the physical tests is strongly questionable. For example, you compare physiological characteristics with technical on-ball attacking and defending indicators. I am quite sure, if you have measured also physiological indicators (e.g. running distance or number of sprinting actions), your correlations in table 3 would increase significantly. Since the investigations have already been completed, this cannot be changed, however, this has to be critically considered in the discussion (e.g. in line 352 ff.).

• I would recommend restructuring your introduction and method parts slightly. First, line 52-56 should be moved to “the current study”. Additionally, line 49-52 should be placed later in the introduction as they provide a conclusion of “Assessment in sports” and “Small-sided games”. Instead you could describe important soccer performance indicators in somewhat more details at the beginning of the introduction. Second, I would not use *sports* in line 57 as your statements about sports in general is limited to only few sentences.

Third, please reconsider your subheadings in material and methods. At this stage, a better distinction between procedures and measures is essential and more clarification of your measured variables is needed. For example, the assessment of physical abilities should not be on another level than the indicators of SSG game performance, both areas are a kind of dependent variables and should therefore appear under the same subheading. Finally, while you try to provide the reader with a comprehensive input on data preparation the important part of statistical analysis section is missing, however, it is vital for better understanding your results.

• After revisiting the material and methods part, the presentation of the results should follow suit. Here, also consistency in labeling your headings would be helpful. From my point of view, line 238-240 belongs to the participant description in the methods part section.

• Regarding the discussion, the utilized method should be more critically reflected (e.g. justify your choice of the individual performance indicators) as should be the previously mentioned rational for comparing physical tests with technical indicators as game performance.

SPECIFIC COMMENTS

Introduction

Title and Abstract

• p.1: based on the general comments an adjustment of the title could be helpful: “The validity of small-sided games in predicting soccer game performance”

• p.2: revise your abstract based on the above recommendations

• p.2, l.41: it is *tests*

Introduction

• p.3, l. 48: point out the different between *characteristics and abilities*

• p.3, l. 62-63: further information is desired on what is meant by *regular game soccer performance*

• p.4, l. 72: *other selection contexts* is not essential, delete it.

• p.4, l. 80-82: It looks like that you chose the weakest median effect sizes in this review with respect to endurance and speed. Exactly as in this study, the authors of the review separated *speed* in different forms (e.g. shorter, longer distances or COD). This should be mentioned here or be taken up again in the discussion

• p. 4, l. 92: Are these *technical abilities* or *technical skills*?

• p. 4, l .92: *r = 0.39* is not a *very strong correlation.

• p.5, l. 106ff: Please consider the general comment of the introduction

Material and Methods

• p.6, l. 121: do you need hyphens in *technical-and-tactical practice”?

• p.6, l. 126: in order to classify better name *national amateur level* in more detail

• p.6, l. 132 ff.: please rephrase this part, currently there are some information making the part confusing, for instance clarify *game day*. Perhaps a small table can provide a better overview?

• p.7, l. 145-146: it is possible that some set of teammates repeat themselves?

• P.7, l. 154-155: reference this typical using of sprint tests [e.g., Altmann, S., Ringhof, S., Neumann, R., Woll, A., & Rumpf, M. C. (2019). Validity and reliability of speed tests used in soccer: A systematic review. PloS one, 14(8)].

• P.8, l. 165: provide a rational why using the T-test

• p.8, l. 181-182: The term *second half* for the season period is confusing here, because you use the same term below to describe the half of a game. Please revise.

• p.9, l. 188: what do you understand by *extensive knowledge*? Please clarify.

• p.9, l. 192: replace or define *events*.

• p.11, l. 213: change the reference and the information in brackets.

Results

• p.12, l. 238-240: I think this belongs to the material and method part, more specifically to participants.

• p.12, l. 245-246: Could you provide a range of your correlations?

• p.13, l. 253-255: The fact that you have removed *aerial duels” before you can just present the final correlation

• p.13., l. 261-271: Think about to place this section in the methods part (e.g. the new “statistical analyses” section)

• p.15, l. 294-297: Please rephrase the part with the sprint distances to clarify the results

Discussion

• p.17., l. 339-351: Please think about the effect that the same experts evaluate the same players both in SSGs and 11 vs.11

• p.18, l. 352-354: see recommendation in general comments

Tables and Figures

Table 1:

• I prefer to place the column *Offensive/Defensive aerial duel* at the end because of your exclusion of these indicators

Table 2:

• Please adapt the statistical *n* for all indicators

Figure 1:

• I would remove the aerial duels in this figure

6. PLOS authors have the option to publish the peer review history of their article (what does this mean?). If published, this will include your full peer review and any attached files.

Reviewer #1: No

Reviewer #2: No

---

## [Author Response · Author response to Decision Letter 0]

10 Jun 2020

Dear dr. Sunderland,

We would like to thank you and the reviewers for the extensive feedback and for highlighting specific areas to be clarified or improved. The reviewers and you have highlighted several issues that helped us to build a stronger and clearer case for the theory and method we employed. Please find our responses to the comments made by you and the reviewers below. 

Editorial comments

1. In your Methods section, please provide additional information about the participant recruitment method and the demographic details of your participants. Please ensure you have provided sufficient details to replicate the analyses such as: a) the recruitment date range (month and year), b) a description of any inclusion/exclusion criteria that were applied to participant recruitment, c) a table of relevant demographic details, d) a statement as to whether your sample can be considered representative of a larger population, e) a description of how participants were recruited, and f) descriptions of where participants were recruited and where the research took place. Moreover, please ensure that the statistical analysis performed has been described in adequate details.

REPLY: We have added additional information about the participants.

A) Recruitment date range is now specified on p. 7-8, L.166 - 169. 

B) A description of inclusion and exclusion criteria is now specified on p.8, L.169 – 174. 

C) We’ve included a table detailing demographic information for the participants per age category. We also added information regarding the average number of small-sided games played in per age category (see p.8, Table 1).

D) We describe more clearly now that our participants can be considered elite youth players in the abstract (see p.2, L.34), methods section (see p.7, L.165) and throughout the manuscript (e.g. p.23, L.483), and have included a statement that our sample can be considered to be representative of the population of elite soccer players in the U15 to U23 age categories on p.8-9, L. 184-186). 

E) We describe more clearly how participants were recruited on p. 7 – 8, L.166-169.

F) We specify where the research took place on p. 9, L.192 – 194 & 201 - 202 

2. Thank you for including your competing interests statement; "The authors have declared that no competing interests exist."

We note that one or more of the authors are employed by a commercial company: Football Club Groningen

2a) Please provide an amended Funding Statement declaring this commercial affiliation, as well as a statement regarding the Role of Funders in your study. If the funding organization did not play a role in the study design, data collection and analysis, decision to publish, or preparation of the manuscript and only provided financial support in the form of authors' salaries and/or research materials, please review your statements relating to the author contributions, and ensure you have specifically and accurately indicated the role(s) that these authors had in your study. You can update author roles in the Author Contributions section of the online submission form.

REPLY: We have updated our funding statement to declare that Football Club Groningen did not play a role in the study design, data collection and analysis, decision to publish, or preparation of the manuscript. Football Club Groningen only provided financial support in the form of salary for one of the authors, and facilitated that the research could be conducted at their club. 

2b). Please also provide an updated Competing Interests Statement declaring this commercial affiliation along with any other relevant declarations relating to employment, consultancy, patents, products in development, or marketed products, etc. 

REPLY: We have included an updated Competing Interests Statements acknowledging the commercial affiliation of one the authors. While this affiliation does not alter our adherence to PLOS ONE policies on sharing data and materials, and while researchers can access our data, we determined some terms of access in consultation with the Ethical Committee Psychology of the University Groningen. These terms are explicitly and clearly explained when clicking the link to our research materials (please see our reply to your comment below).

REPLY: We agree that we did not provide sufficient information regarding the data availability statement. Before our first submission, we had uploaded our data and analyses on Dataverse, the link of which was (and is) added in the manuscript. Based on potential issues with identifiability of participants, we discussed a procedure with the ethical committee in which our data can be shared with, and used and analyzed by, other researchers. We will clarify this point in more detail:

With respect to data sharing, the data itself contains information that could be identifiable given the identification of the club involved (by deducing from the author affiliations), publicly available information (e.g., team age categories) and the relatively small sample in the study. In consultation with the Ethical Committee Psychology of the University of Groningen, restrictions on openly sharing this data were applied. However, the data is deposited at a Dataverse repository that allows for controlled data access to all interested and qualifying researchers: https://hdl.handle.net/10411/XEVAVU. Data requests can be also be send to j.m.baan@rug.nl, who is a non-author institutional point of contact. All this information, along with how to use and analyze the data, can be found when clicking the link (the link is activated upon acceptance of the manuscript).

Reviewer #1

GENERAL COMMENTS:

Thank you for your contribution to PLoS One. Overall, the paper which presents a study analysing the validity of performance characteristics of small-sided games for soccer performance within young soccer players falls within the scope of the journal and should be of interest to the readership. The study’s purpose was to validate different performance indicators assessed during small-sided games (in particular technical/tactical indicators) or physical/motor tests (e.g., linear sprint performance, endurance) to predict technical/tactical performance characteristics in 11 vs 11 “real” matches.

Strengths of the study mainly relate to the idea of using continuous soccer performance outcomes of competition matches to measure success in soccer, the attempt to develop a performance score based on various soccer outcomes (even though they include at least in parts subjective observations) as well as the idea to consider game-specific measures (SSG) to predict soccer performance. However, in its current form the manuscript also presents a range of concerns in regard to its content (aims of the study, utilized predictors/criterion) and presentation (particularly structure and description of methodology).

 REPLY: Thank you for your evaluation of our manuscript and for your helpful suggestions. Below you can find our answers to your comments.

1. First, please consider (re-)defining the aims of your study more precisely and reflect on your title. In my point of view, purpose 1 (representativity of SSG for 11 vs 11) and purpose 2 (validity of SSG performance in predicting regular game performance) are quite similar. The real distinction between those two is not precisely explained. Consider combining these ideas as they are rather related.

 REPLY: While we understand your suggestion, we see these two aims as distinctive. Our argument is as follows: our first aim was to assess the degree to which 7-vs-7 SSGs can be considered as a representative assessment context for competitive 11-vs-11 games. Specifically, we aimed to evaluate to what extent the two game formats resemble each other, in terms of the distribution of the observed events on the different performance indicators (on a team level). From this perspective, this analysis can be considered to assess the content validity of the SSGs. Our second aim was to assess whether individual performance on the indicators is related to performance of the individual players in 11-vs-11 games. Thus, in our second aim we focus the predictive validity of individual SSG performance. 

In the revised manuscript, we describe the distinction between the two aims more clearly, and have revised part of our introduction to provide a more accurate lead-up to these two different aims. Specifically, we describe in more detail that the degree of representativeness of SSGs may be dependent on variations in the specific number of players and pitch size, and that it is important to assess this level of representativeness (see p. 6, L.129-131). In addition, we have defined our aims more precisely in section ‘The current study,’ p. 7, L.150 – 157. 

2. Much more important are my concerns with regard to purpose 3 (validity of physical tests for in-game soccer performance). On the one hand I see an issue with your terminology: as you use sprint, endurance as well as agility tests here, you cannot use “physical tests” for your selection of measures (physiological parts as well as technical skills and speed abilities are included).

 REPLY: We agree that ‘physical tests’ incorrectly described the tests used our study or provided a too narrow description of the tests used. We have changed physical tests to ‘physiological and motor tests’ throughout the manuscript and in the title (see for example p.10, L. 228, section ‘Predictor: physiological and motor tests’).

3. Further, please provide a rational for your choice. Why did you use these three tests (and not others)?

 REPLY: We aimed to assess the predictive validity of separate tests that are often conducted in the soccer field and are often described in the talent literature (e.g., Murr et al. 2018). The three reported tests are also typically conducted at the club where we did our research, and many other clubs in our country. Moreover, they are relatively straightforward to assess with acceptable levels of reliability (e.g. Lemmink et al. 2014; Haj-Sassi et al. 2011). In previous literature, tests taken at soccer clubs are usually included as predictors for whether players are selected or not, are elite or non-elite, etc. We now specify in some more detail why we included these tests (see p.7, 157 - 162). 

4. However, I am even more concerned about the general idea/procedure of how you deal with purpose 3. Physiological as well as speed related test performances are assessed and correlated with mainly specific technical on-ball performance indicators (also given as a limitation of your study in l.352 pp). In my point of view, this explains the quite low correlations between your tests and the in-game performance indicators. Therefore, this should not lead to the conclusion that your used tests are not valid to predict soccer performance. In my point of, you should either use other parameters (physiological, physical) on the “criterion level” or (if there is no possibility to assess such data) think about revising/removing this purpose. In every case you should discuss more precisely that there is also validity given to individual physical (and others) tests to predict soccer performance in the introduction and/or discussion. There is various current literature dealing with this (e.g. Emmonds et al. (2016), Höner et al. (2017), Johnston et al. (2018), …). .

REPLY: We understand your concerns. In the revised manuscript, we have dropped purpose 3 as an aim of the study (see p.7, ‘The current study’), but we kept the correlation between these tests and in-game performance as an exploratory part of the study. In addition, we provided a more nuanced discussion of the relationship between these tests and our operationalization of in-game performance. 

The reason why we kept this relationship in the paper is as follows. An important aim of this study was to incorporate a continuous measure of in-game soccer performance, given recent criticisms of the dichotomization of the criterion into performance levels. In order to quantify in-game performance, we used notational analysis to assess on-ball performance indicators. We used these indicators, because recent work suggests they are related to game success (i.e., winning, see Pappalardo et al. 2017). In contrast, evidence for the relationship between physiological in-game parameters (e.g., total distance run) and game success has been mixed (e.g. Hoppe et al., 2015). We therefore considered it relevant to assess how tests commonly used in the talent field relate to this continuous in-game performance outcome. 

Nevertheless, please note that we agree that the low correlations between the physiological tests and our operationalization of in-game performance are quite logical. They might be expected based on the perspective that the physiological tests and the performance indicators have relatively little in common in terms behavior and context. In the revised manuscript we explicitly acknowledge this (e.g., see p.6, L.122 - 126; p.7 L.160 - 162; p.22, L.466). In addition, we elaborate on our choice to still look at the predictive validity of these tests, as well as how it this choice fits within the general purpose of the article and framework of behavioral consistency (see revised introduction; e.g. p.5-6, L.117 - 126). 

Finally, we acknowledge that there are studies which have found positive associations between physiological tests and performance level (see p. 5, L.98 - 104), and specifically state that our findings on the predictor-criterion relationship for the physiological tests hold true for the performance indicators that we chose (see p.20, L.409-412; p.23, L.464 - 469). 

5. At the same time, the even higher correlations between SSG and 11 vs 11 performance (almost at the same time point) can be explained: same measures used for SSG and 11-vs 11 protocol which is ok for the analyses of representativeness, but not for a comparison between validity of SSG performance and physical tests in order to predict in-game (technical) soccer performance. Please double-check.

 REPLY: We agree. We explain more clearly why, from the perspective of behavioral consistency, relatively high correlations for the performance indicators in the SSGs and 11-vs-11 games may be expected. That is, these correlations make sense, because the predictor mimics the criterion behavior in content and context. The purpose of aim 2 was to examine how well this notion holds up for soccer, by examining the predictive validity of SSG performance in relation to performance in 11-vs-11 games. We describe this line of thinking more clearly now in the introduction of the revised manuscript (see p.5-6, from L.117), and reflect more on how these findings match our expectation in the discussion (p.22-23, from L.462). 

 Also, in line with the idea of behavioral consistency, the relatively low predictive validities of the physiological and motor (i.e., low-fidelity) tests could be expected. As noted, we have placed less emphasis on this comparison in the revised manuscript. Specifically, we have revised our aims (see p.7, ‘The current study’) and do no longer state this comparison as a main objective. Accordingly, we also provide a more nuanced discussion of this comparison in the revised discussion section and conclusion (see p. 25, L.522 - 524). 

6. Further, there are a few issues with the description of the methodological process (see also specific comments and general comment with regard to purpose 3). Your materials and methods section has some inconsistencies with regard to structure. Although I like the idea of using various subheadings to generate a structured section, the authors should reflect this choice. As you describe in your “procedures” the way you proceeded with the SSG, physical tests and 11-vs-11 games, I was quite surprised that there was a Measures section in the following where the notational analysis and the score development war described. In my point of view, this information belongs together and should not be separated (perhaps in one “measures”-section with subheadings divided by dependent and criterion variables?).

 REPLY: In line with your comment, we have solved the inconsistencies in the structure of the methods section in the following ways:

• We have changed the subheading ‘procedure’ to ‘procedure and measures.’ (see p.9, L.190)

• We clearly distinguish between predictor and criterion data in this section, adding ‘predictor’ before the subheading of the SSG section and physical test section, and ‘criterion’ before the subheading of the 11-vs-11 games section (see L.191, L.228, and L.257)

• We have combined the ‘notational analysis’ section with the ‘procedures and measures’ section. Specifically, we have added the information to the section describing the SSG protocol (see p.10, from L.221). In addition, we added a small paragraph at the end of the 11-vs-11 games section stating that we used the same notational analysis procedure for the SSGs to code performance in the 11-vs-11 games (see p.12, L.270).

• We added a separate statistical analyses section after the ‘data preparation’ section. Descriptions of the statistical analyses at the start of each result subsection have been moved to this section (see p.15, L.316). 

• We moved information on the participants who were excluded from certain analyses to the ‘statistical analysis’ section (see p.16, L.333 – 339). This information was originally placed at the start of the results section. 

We hope that these changes have clarified our methods section, while also making the steps taken in our analytic procedure easier to follow. 

7. By the way: you sometimes use 11-vs 11 games, sometimes in-game performance: please be consistent, and more important: define, in the introduction, what you mean by in-game performance.

REPLY: We now use ‘in-game’ performance throughout the manuscript to refer to measures of performance (e.g., performance indicators) within the SSGs and 11-vs-11 games. Moreover, we refer to 11-vs-11 games, instead of regular games or regular game performance, when discussing the criterion context. Additionally, and more importantly, we agree that a discussion of what in-game performance means was missing from our manuscript. Indeed, in-game soccer performance comprises the interaction between technical, tactical, physiological, and perceptual-cognitive behaviors, and there are various methods to assess these different aspects of in-game performance (e.g., GPS to assess meters ran, LPS to assess spatio-temporal information, notational analysis to assess technical and tactical characteristics). We discuss this early in the revised introduction, and motivate our choice for the performance indicators we chose (see p.3-4, ‘Soccer performance criterion, from L.65). Finally, we also reflect more critically on our chosen performance indicators in the limitation section (p.24, L.490 – 503).

8. Furthermore, there is a statistical analysis section missing. As you partly describe which statistical measures were used in the results (and in the notation analysis) section, it is hard to understand for the readership what you exactly did. Please provide a separate section where you describe your analyses in more detail (especially provide rational why you use aggregated rank correlations, see also specific comments on Figure as well as purpose 1.)

 REPLY: We have added a separate ‘statistical analysis’ section (see our reply to general comment #6; see p.15, L.316), where we describe our rationale for the use of Spearman’s rank correlation to assess the association between the distributions (see also specific comment #38). 

Specific comments

Title

1. Depending on your decision regarding general comments: think about removing the physical tests from the title

 REPLY: We have changed the title of the manuscript to: the validity of small-sided games in predicting 11-vs-11 soccer game performance.

Abstract

2. L. 25: delete redundancy (research)

 REPLY: This has been addressed (see p.2, L.24).

3. L. 27: representative for? Please incorporate.

REPLY: This sentence has been removed in the revised abstract (see p.2, L.24-29).

4. L.29ff: revise, combine purposes (see general comment)

 REPLY: we did not combine the purposes, but have aimed to describe the difference between them more clearly (see also our reply to your general comment #1)

5. L.33: “in order to”?

 REPLY: This sentence has been removed in the revised abstract.

6. L.41: “tests”

 REPLY: Thanks for pointing out this typo. This has been addressed (See p.2 L.42).

Introduction

7. L.49: Please elaborate and add some information about studies dealing with this.

REPLY: We discuss these studies now in some more detail (see p.5, from L.98). 

8. L.52-56: Please move this part to “The current study” and combine it with the information given there.

 REPLY: this has been addressed. We have combined it with the information in ‘The current study’ in the revised manuscript (see p.7). 

9. L.63: please define regular game performance (see also general comment)

REPLY: This sentence has been removed in the revised manuscript. We agree that regular game performance was the incorrect terminology here. We now use ‘performance in 11-vs-11 games,’ or ‘in-game performance,’ and define in-game performance in the introduction (see our reply to your general comment #7).

10. L.76-79: provide reference/evidence

 REPLY: we have provided some references here to support this statement (see p.6, L.123). 

11. In my point of view, the prognostic value could not be described as “relatively weak” in this case, as there are medium effect sizes (and various studies I am aware of which state that there is an evidence!). Please revise and add more information (here or in the discussion).

 REPLY: We meant to refer here to the median effect sizes for speed (<20m), agility, and endurance capacity across the studies addressed in the systematic review, which aimed to synthesize the evidence (Murr et al. 2018). These can be considered small, based on the guidelines specified by Cohen (1988). However, we agree that we there are physiological variables in the systematic review for which medium and large effect sizes were found, and which we didn’t report in the manuscript. We acknowledge these effect sizes in the revised manuscript. Furthermore, we have rephrased this sentence, and do not longer describe the prognostic value as ‘relatively weak.’ Instead, we state that these studies had various levels of success in discriminating between performance levels (see p.5, from L.98-99).

12. L.92: ability? Or skills?

 REPLY: we have changed ‘technical ability’ to ‘technical skills’ here (see p.6, L.139). 

13. L.92: “very strong” is not the right terminology for a correlation of r = 0.39. Please revise.

 REPLY: we agree, we have changed the ‘very strong relationships’ to ‘strong to moderate relationships’ (see p.6, L.138). 

14. L.94: ad “significantly”?

 REPLY: This has been addressed, we have added ‘significantly.’ (see p.6, L.141)

15. L.100ff: provide reference

REPLY: We have added some references to support this statement (see p.7, L.146).

16. please revise and combine with L.52-56 (please, avoid redundancies)

 REPLY: This has been addressed (see specific comment #8).

Methods

17. L.119: Add „N = 63“.

 REPLY: This has been addressed (see p. 8, L.174). 

18. L.122: Did you control for the different performance levels of the teams? Please provide information and/or discuss.

 REPLY: We controlled for differences in performance level in our analysis of the predictor-criterion relationships, by computing the correlations within each team first, and then combining the correlation coefficients through a random effects meta-analysis. By doing this, we avoid comparing players of different age categories directly, such as comparing U15 players against U19 players. We now describe this more clearly in the statistical analysis section (See p.16, L.340 - 346). Moreover, the SSGs were played by each age category separately, meaning that we did not ‘mix’ players across the different age categories in the SSGs. 

19. L.132 ff. pleas revise as this part remained unclear.

 REPLY: We have revised this paragraph to more adequately describe that the number of SSGs played within participants and across teams was not exactly the same, due to uncontrollable circumstances (see p.9, L.169 - 200). In line with editor comment #1, we have also added a table detailing the average number of SSGs players in each age category participated in (see p.8, Table 1). 

20. L.148: “two months after the beginning of the season”?

 REPLY: We agree, this has been addressed (see p.10, L.229 - 230).

21. L.149: “linear” instead of straight line?

REPLY: This has been addressed (see p.10, L.231).

22. L.155: delete “the” and use “each athlete” to avoid redundancy.

 REPLY: We agree, this has been addressed (see p.11, L.236).

23. L.165: provide rational for the use of this test to measure agility (as far as I know, this test is not specific to soccer) (see also general comment)

 REPLY: We included this test (and the other two tests) tests because they have been used in soccer research (e.g., Altman et al. 2019, Lemmink et al, Sporis et al., 2010, Murr et al.,2018), and are frequently used by soccer clubs to monitor or predict performance (including the club at which the current research project took place; see our reply to your specific comment #3). The T-test is a slightly adapted version of the test specified by Haj-Sassi et al. (2011), by making players sprint around cones, instead of shuffling sideways (see p.11, L.252 -253). 

24. L.175: delete “the”

REPLY: This has been addressed (see p. 12, L.258).

25. L.176: “that were filmed”? Please revise.

REPLY: This has been addressed (see p.12, L.259-260).

26. L.180ff: provide a rational for the use of only one half of the match.

 REPLY: We aimed to match the total number of minutes in the SSGs to the number of minutes in the 11-vs-11 games. This corresponded to analyzing approximately three full 11-vs-11 games for each age category. However, we instead opted to use one half of 6 different matches, to obtain more variability in opponent strength, as well as individual player variability across games. We now mention this in the text (see p.12, L.260-264). 

27. L.184: think about combining with procedures (see also general comment)

REPLY: Agree, we have combined this with procedures (see general comment #6).

28. L.189: please provide information about the validity of the utilized coding scheme (level of experts, procedure, …)

 REPLY: We now explain that the coding scheme was developed in consultation with the club’s head of performance and data analyst. They each have > 7 years of experience with soccer event data (see p.10, L.212-226). We have also provided additional information on the coding process conducted by the researcher and students (p.10, L.223-226).

29. L.192: Can you provide some examples for events in SSGs here?

 REPLY: We refer to ‘events’ simply as the observations/observed actions on a performance indicator, regardless of outcome (e.g. total number of passes, regardless of whether they were (un)successful). This is in line with soccer analytics research, where this type of data is often referred to as event data (see for example Brooks et al., 2016). We’ve added a sentence to specify this (see p.10, L.215 - 216).

Since we provide examples of the performance indicator in the paragraph prior to L.192, we do not list examples here. However, we have changed ‘events’ to ‘performance indicators’ to indicate more clearly what we refer to in this sentence (see p.10, 221). 

30. L.196: Why only 3 of the 6 matches were rated by 3(?) persons and what happened to the other 3? Provide information.

REPLY: We agree that this was confusing. We meant to specify that we coded 3 of the in total k = 82 SSGs to assess the inter-rater reliability. Based on these 3 SSGs, we computed the inter-rater reliability. We now describe this more clearly here (see p.10, L.225 – 226). In addition, we provide some more details on the coding procedure here (p.10, L.223-224).

31. Table 1: Definition Offensive aerial duel: add “ball” to possession.

REPLY: This has been addressed (see Table 2, p. 13).

32. Table 1: a definition of “shots on target conceded” is missing here. Further, why did you not include this indicator with weight 1 in the Defense-score as you did for offense? Please, explain.

REPLY: We refer to shots on target conceded as the number of shots on target taken by the opposite team. Thus, if team A had 3 shots on target, team B had 3 shots on target conceded. Since we coded events for two teams for each SSG played at the academy, the shots on target conceded per team, per SSG, could be easily obtained. For the competitive 11-vs-11 games, this meant that we also coded the shots on target for the opposite team; a team which was not part of the academy. The shots on target (conceded) per game (i.e., SSG and 11-vs-11) were used to compute the weights for the offensive and defensive performance composite metrics, to assess the relationship between the performance indicators and offensive and defensive success, respectively. However, while shots on target conceded could be obtained for a team, they are not bound to one player, since a single player does not specifically ‘allow’ a shot on target. This is in contrast to shots on target, which is available as a statistic for both a team and individual player. Hence, we could not give shots on target conceded a weight of 1, as we could not assess as a defensive indicator per player (e.g., see Pappalardo et al. 2019). We now specify that shots on target conceded are the shots on target for the opposite team in the text (see p.14, L.302). 

33. L.225/226: provide rational for the assessed indicators (see also notational analysis and general comment)

 REPLY: We now describe more precisely why we chose these indicators here (see p.10, L.212-220), as well as in the introduction (see p. 3-4, ‘Soccer performance criterion’). Moreover, we critically reflect on our chosen indicators in the strengths and limitations section of the revised manuscript (p.24, L.490 – 503).

34. L.233: did you compute the correlations team-wise as well? Or in general (what I would guess). Did you check whether there are big differences with regard to the correlations?

REPLY: The correlations were computed team-wise, per game format. To control for differences in performance level, as well as the number of games, we aggregated these correlations using the meta-analysis method that was also used for the predictor-criterion analyses. The largest differences were in the 11-vs-11 games. However, it should be noted that these correlation estimates are based on a smaller sample size (i.e., six 11-vs-11 halves per team), and are therefore more unstable than the estimates based on the SSGs. We have included all correlations as a table in the supporting information (see S3 Table). Moreover, we have added a sentence to describe more clearly how we derived the correlation in table 2 (see p. 14-15, L.305 - 307).

35. L: 238-240: this information should be part of the method section. By the way, why not remove the players who were excluded from analysis from the study sample?

REPLY: We have moved this information to the methods section (see p.16, L.333-339). We have decided to include players with missing data on the physiological tests, but with sufficient data in the SSGs and 11-vs-11 games, in the SSG predictor-criterion analyses. Excluding these players from the analysis would mean we applied listwise deletion across the statistical analyses, which is generally discouraged in the literature (see Harel et al, 2008). 

Results:

37. Present statistical analysis in the method section separately (see general comment)

REPLY: This has been addressed (see general comment #8 and p. 15)

38. L.244: I struggle if Spearmans rank correlation is the right measure here. I would have expected t-Tests/non parametric adequate here to provide information about differences within the frequencies of events (for single indicators and scores). In doing so, you get information about each individual indicator as well as your score, no?

 REPLY: We considered Spearman’s rank correlation as the right measure. At the same time, your comment made us realize that we could provide a more rigorous analysis to assess the degree of representativity of the SSGs, compared to 11-vs-11 games. 

To assess the degree of representativity of the SSGs, Fig 1 now displays the mean number of events p. 6 minutes for each performance indicator, for both game formats (see Fig 1). We have added a table in the in the supporting information detailing this information as well (see S4 Table). Furthermore, we added a chi-square goodness of fit test the observed events in the SSGs per performance indicator (i.e. the empirical distribution) against the distribution of events in the 11-vs-11 (the theoretical distribution) (see p. 15, from L.317 & p. 17, from L.362). In line with your suggestion, we obtain information on each individual indicator through these analyses. Because effect size estimates for chi-square can be difficult to interpret, we conduct a spearman’s rank correlation on the total number of observed events in both game formats, to assess the association between the distributions. We elaborate on this choice in the statistical methods section (see p.15, L.326-328).

39. L.256: Not clear what this means. Please provide more information. Why did you aggregate team-wise? Is there no influence of age class?

 REPLY: We agree that the term ‘aggregate’ in this sentence was confusing, given that we also discuss aggregated correlation coefficient resulting from the meta-analyses. This sentence has been removed in the revised manuscript (see p.17-18, L.374-377).

40. Figure 1: it would be beneficial to add some inferential statistics here as well (e.g.: results from test of mean/median differences?)

 REPLY: We have changed Fig 1 to include the mean events p. 6 minutes in both game formats. This allows us to plot the variance/spread around and obtain information on each performance indicator separately (see Fig 1, see also reply to specific comment #38). In addition, we report the confidence intervals around the means in the table in the supporting information (see S4 Table). However, we’ve decided not to report or analyze p-values here. Our argument is that the mean events in the SSGs are based on many more datapoints (k = 82*2 = 164, since we analyze both teams), than the 11-vs-11 games (k = 24). Given that p-values are dependent on the sample size, and do not provide accurate information about the strength of the effect, we do not consider tests on these means for each performance indicator as meaningful here. 

41. Figure 1: provide more information about the different categories: are the shots not part of chances created? If so, I wonder why the frequencies of shots in 7-7 is higher than for chances created in 7-7?

 REPLY: the different categories reflect the different performance indicators used throughout the manuscript. A brief definition of each performance indicator is provided in Table 2, and we provide a more elaborate definition in S1 (supporting information). Chances created refers to the combination of assists and key passes. While these are related to shots, not every shot is the result of an assist or key pass. 

 Additionally, we noticed that we used ‘shots’ in Fig 1, instead of shots on target. We have changed this to shots on target, to more accurately correspond to the performance indicators used in the rest of the manuscript (see Fig 1).

42. L.265: “59 <= N <= 63”?

REPLY: This has been addressed (see p.16, L.342).

43. L.266: Please provide more information about the aggregation of coefficients (see also general comment)

 REPLY: In order to control for differences in performance levels, we first computed correlations within each age category. Then, we combined these correlation coefficients through a random effects meta-analysis. This method accounts for potential heterogeneity in the coefficients, and results in a weighted average correlation. We refer to this weighted average correlation coefficients as the aggregated coefficient in the results. We have described this analysis in some more detail in the statistical analysis section (see p. 16, from L.340 - 346). 

44. L266-271: move this part to the statistical analyses section.

REPLY: This has been addressed (See p.16, from L. 329).

45. L.272: “aggregated”?

REPLY: This has been addressed (see p. 18, L.380).

46. L.273ff: remove redundancy.

REPLY: Agreed, we have removed this sentence. 

47. L.285-286: I think this result should be discussed in detail within the discussion section (maybe also by giving perspectives for future research as well as dealing with the limits of the used statistical procedures (correlations). 

 REPLY: we have elaborated on these findings in the discussion (p.22, L.453 – 459). Moreover, we discuss some perspectives for future research more explicitly in the conclusion (see p.22, L.529 – 534).

48. Table 2: provide n in every row

REPLY: This has been addressed (see p.19, Table 4 and p.20, Table 5).

49. Table 2: would it not be beneficial to present correlations coefficients for each age class as well?

REPLY: We controlled for differences in performance level by computing the correlations within each age category. However, given that each correlation per team is based on a small sample size (e.g. 13 to 17 players), we do not mean to place too much emphasis (in terms of inference or comparisons) on each coefficient individually. Although we present the correlations for each age class in the supporting information (see S5 Table and S6 Table) for completeness, we decided to not present all of them in the manuscript text and overanalyze the single coefficients. 

50. L.288 pp: see general comment

 REPLY: this has been addressed, we’ve changed this title to ‘physiological and motor tests’ (see p.10, L. 228)

51. L.293/294: “left” and “right” instead of top/bottom.

REPLY: Thanks for noticing this error, this has been addressed (see p.19, L.400 - 401).

52. L.295: “effectively zero”? check wording.

REPLY: We have changed this to ‘trivial’ in the revised manuscript (p.20, L.402.)

53. L.295pp provide p-values/CIs/information about significance here.

REPLY: this has been addressed, we have added CIs here (see p.20, L.402 – 408, and p. 18-19, L.385 – 392)

54. Table 3: “p” instead of “p-value”?; provide n in every row.

REPLY: This has been addressed (see p.20, Table 5).

Discussion

55. L.304: a discussion about the right measure of “regular game performance” is needed.

 REPLY: We agree, see our reply to general comment #7. We reflect more critically on our chosen performance indicators in the ‘strengths and limitations’ section (p.24, L.490 – 503).

56. L.306pp: please note, that this only holds true for the indicators you chose (technical, on-ball) during real matches. Please consider and add information

 REPLY: we now explicitly state throughout the discussion section that the degree of representativeness, as well as the predictive validity, hold true for the indicators we chose. (e.g., see p.20, L.411 – 412; p. 23, L.465 - 466, and L.468-469)

57. L.306pp: please also reflect on the fact, that your correlations could also be “that high” because you assessed SSG an 11-11 with the same measures within a quite short time window.

 REPLY: We now discuss that the assessment of performance indicators in the short time window could have affected the correlations (see p.24, from L.509 - 511).

58. L.326-329: Is there further evidence? A more nuanced discussion would add to the content here.

REPLY: While there are various studies that have related the physiological tests to performance level (as noted in our introduction), we are not aware of any other studies that relate these tests to a continuous measure of in-game performance, as assessed through notational analysis. Therefore, we were unable to discuss similar studies in this section. No changes in text.

59. L.330-L337: Although I see and like the authors purpose of this paragraph, I think that it should be extended (as it represents a main issue and at the same time limitation of the studs, see general comment).

 REPLY: We have extended the paragraph by adding the following information: “This means that physical performance is most likely related to soccer performance in the general population of all youth players. However, there is not enough variance in physical performance between elite soccer players to meaningfully differentiate between them, as it is possible that the elite players have, explicitly or implicitly, been preselected on these variables” (see p.23, L. 490 - 484).

60. L.352 pp: see general comment

REPLY: We agree, see our reply to your general comment #2

61. L.360: please revise according to your decision about your aims and also give some perspectives for future research. 

 REPLY: see our reply to your general comment #4 and #5. Furthermore, we’ve added some perspectives on future research here (see p.26, L.525 – 530).

References not in manuscript

Brooks, J., Kerr, M., & Guttag, J. (2016). Developing a Data-Driven Player Ranking in Soccer Using Predictive Model Weights. Proceedings of the 22nd ACM SIGKDD International Conference on Knowledge Discovery and Data Mining - KDD ’16, 49–55. https://doi.org/10.1145/2939672.2939695

Harel, O., Zimmerman, R., & Dekhtyar, O. (2008). Approaches to the handling of missing data in communication research. The SAGE sourcebook of advanced data analysis methods for communication research, 349-371.

Reviewer #2

Decision: Major revision

The current manuscript is well-written and covers a topic which should be of interest to other researchers within the field of talent identification and development. The presented study investigates the predictive validity and impact of small-sided games (SSG) on soccer performance. Furthermore, the approach of examining the predictive validity of small-sided game performance is novel and addresses a range of previously identified methodological issues. However, at times a lack of clarity with respect to the presented content makes the manuscript hard to follow and therefore needs to be addressed before potentially considering this paper for publication. Thus, in the following, you will find some general and specific recommendations which are thought to help improving this paper:

REPLY: Thank you for your interest and evaluation our manuscript. Your suggestions have certainly improved the clarity and overall quality of our manuscript. There was substantial overlap between your suggestions and those of reviewer #1. Therefore, we have often provided a brief answer to your suggestions, and refer specifically to our replies to reviewer #1 comments, to avoid redundancies. 

GENERAL COMMENTS

1. In general, a more consistent use of terminology throughout the entire manuscript is necessary. For instance, better use the same term for your criterion (i.e. in-game performance vs. regular soccer game performance vs. performance in 11 vs. 11 games).

 REPLY: We agree. We now use ‘in-game’ performance to refer to performance indicators in the SSGs and 11-vs-11 games. Moreover, we use 11-vs-11 games, instead of ‘regular games’ to refer to the criterion (11-vs-11) context (See our reply to reviewer #1, general comment #7.)

2. With respect to your proposed criticisms it is not clearly stated what added value the first one has over the second. My suggestion would be to combine the two and thus only state 2 objectives. 

REPLY: We consider the first aim to assess the content validity of SSGs in relation to 11-vs-11 games, whereas the second aim addresses the predictive validity of SSG performance. Therefore, we see these aims as distinctive. We have aimed to describe the difference between the aims more clearly. See ‘The current study,’ p. 7, L.150 – 157, and our reply to reviewer #1, general comment #1. 

3. In addition, while your methods of validating the SSG is well thought through and practical, your way of examining the predictive validity of the physical tests is strongly questionable. For example, you compare physiological characteristics with technical on-ball attacking and defending indicators. I am quite sure, if you have measured also physiological indicators (e.g. running distance or number of sprinting actions), your correlations in table 3 would increase significantly. Since the investigations have already been completed, this cannot be changed, however, this has to be critically considered in the discussion (e.g. in line 352 ff.).

 REPLY: We have addressed this comment in three ways. First, we have dropped our purpose of comparing the predictive validities as a main objective of the study (see p.7, ‘The current study’) Second, we provide a more nuanced discussion of the predictive validity of these physiological tests throughout the revised manuscript (e.g., see p.6, L.122 - 126; p.7 L.160 - 162; p.22, L.466; p. 25, L.522 - 524). Moreover, we reflect more critically on our operationalization of in-game performance in the revised introduction and discussion section (see p.3-4, ‘Soccer performance criterion, from L.65; p.24, L.490 – 503 and see our reply to reviewer 1, general comment #4 and #5). 

4. I would recommend restructuring your introduction and method parts slightly. First, line 52-56 should be moved to “the current study”. Additionally, line 49-52 should be placed later in the introduction as they provide a conclusion of “Assessment in sports” and “Small-sided games”. Instead you could describe important soccer performance indicators in somewhat more details at the beginning of the introduction. Second, I would not use *sports* in line 57 as your statements about sports in general is limited to only few sentences. 

REPLY: We have restructured our introduction and method section to address these issues and avoid redundancies. We have described important predictors that have been explored in the talent ID literature in more detail here (see p.5, from L.98). Finally, we have changed ‘assessments in sports’ to ‘assessments in soccer’ (see p.5, L.97). See also our reply to reviewer 1, general comment #6 and #8. 

5. Third, please reconsider your subheadings in material and methods. At this stage, a better distinction between procedures and measures is essential and more clarification of your measured variables is needed. For example, the assessment of physical abilities should not be on another level than the indicators of SSG game performance, both areas are a kind of dependent variables and should therefore appear under the same subheading. Finally, while you try to provide the reader with a comprehensive input on data preparation the important part of statistical analysis section is missing, however, it is vital for better understanding your results.

 REPLY: We have changed the subheading ‘procedure’ to ‘procedure and measures’ (see p.9, L.190) and have combined the notational analysis section with the SSG section (see p.10, from L.221). In addition, we have included a statistical analysis section, and moved the information on the analyses from the results to this section (see p.15, L.316, and our response to reviewer 1, general comment #6, and specific comments #8 and #35).

5. After revisiting the material and methods part, the presentation of the results should follow suit. Here, also consistency in labeling your headings would be helpful. From my point of view, line 238-240 belongs to the participant description in the methods part section.

 REPLY: this has been addressed (see p.16, L.333 – 339 and our reply to your comments above, as well as our response to reviewer 1 specific comment #35).

6. Regarding the discussion, the utilized method should be more critically reflected (e.g. justify your choice of the individual performance indicators) as should be the previously mentioned rational for comparing physical tests with technical indicators as game performance.

 REPLY: We elaborate on our chosen performance indicators in the introduction and methods section (see p.5, L.122 – 126; p. 3-4, ‘Soccer performance criterion’; p.10, L.212 - 220). Moreover, we critically reflect on our operationalization of in-game performance in the ‘strengths and limitations’ section (see p.24, L.490 – 503, and our response to reviewer 1, general comment #4 and 7.

SPECIFIC COMMENTS

Introduction

Title and Abstract

1. p.1: based on the general comments an adjustment of the title could be helpful: “The validity of small-sided games in predicting soccer game performance”

 REPLY: We have adjusted our title to: The Validity of Small-Sided Games in Predicting 11-vs-11 Soccer Game Performance

2. p.2: revise your abstract based on the above recommendations

 REPLY: this has been addressed, see abstract.

3. p.2, l.41: it is *tests*

REPLY: Thanks for pointing out this typo. This has been addressed (See p.2 L.42).

Introduction

4. p.3, l. 48: point out the different between *characteristics and abilities*

REPLY: We agree that these concepts are different. Instead of going into the discussion of the difference in the manuscript, we decided to replace these concepts by the term “attributes”, as has often been done in previous research as well (Larkin and O’Connor, 2017; Roberts et al. 2019), Moreover, we list some examples of the attributes that have been used on soccer TID in the introduction (see p.3, from L. 52). 

5. p.3, l. 62-63: further information is desired on what is meant by *regular game soccer performance*

 REPLY: We agree, we now provide this discussion early in the introduction, and motivate our choice for the assessed performance indicators here. Moreover, we critically reflect on our operationalization of soccer performance in the limitations section. See p.24, L.490 – 503 and our response to reviewer #1, general comment #7

6. p.4, l. 72: *other selection contexts* is not essential, delete it.

 REPLY: This has been addressed (see p. 6, L.120). 

7. p.4, l. 80-82: It looks like that you chose the weakest median effect sizes in this review with respect to endurance and speed. Exactly as in this study, the authors of the review separated *speed* in different forms (e.g. shorter, longer distances or COD). This should be mentioned here or be taken up again in the discussion.

 REPLY: We agree, we now discuss findings on other physiological performance measures discussed in the review (See p.5, L. 98 – 104, and our reply to reviewer 1, specific comment #11).

8. p. 4, l. 92: Are these *technical abilities* or *technical skills*?

 REPLY: we have changed this to technical skills (see p.6, L.138). 

9. p. 4, l .92: *r = 0.39* is not a *very strong correlation.

REPLY: we agree, we have changed the ‘very strong relationships’ to ‘strong to moderate relationships’ (see p.6, L.138). 

10. p.5, l. 106ff: Please consider the general comment of the introduction

 REPLY: This has been addressed (see our reply to your general comment #4)

Material and Methods

11. p.6, l. 121: do you need hyphens in *technical-and-tactical practice”?

REPLY: this has been addressed (see p. 8, L. 178 – 179).

12. p.6, l. 126: in order to classify better name *national amateur level* in more detail

 REPLY: we have changed this to highest adult amateur level (p.8, L.183)

13. p.6, l. 132 ff.: please rephrase this part, currently there are some information making the part confusing, for instance clarify *game day*. Perhaps a small table can provide a better overview?

REPLY: This has been addressed, we have changed game day to ‘practice session’ (see p.9 L.195). Moreover, we have included a table with descriptive information of the participants (see p. 8, Table 1)

14. p.7, l. 145-146: it is possible that some set of teammates repeat themselves?

 REPLY: In theory this is possible, but it is highly unlikely. A quick, back of the envelope calculation suggests that there are approximately 12,600 combinations for a team. 

15. P.7, l. 154-155: reference this typical using of sprint tests [e.g., Altmann, S., Ringhof, S., Neumann, R., Woll, A., & Rumpf, M. C. (2019). Validity and reliability of speed tests used in soccer: A systematic review. PloS one, 14(8)].

REPLY: we now refer to this paper at p.11, L.234

16. P.8, l. 165: provide a rational why using the T-test

REPLY: While this test, and other tests, are not specific to soccer, we included these tests because they are often conducted in soccer science and practice. The reported t-test is a modified version of the t-test specified by Haj-Sassi. We adapted this test to better suit soccer players, by making players sprint around cones, instead of shuffling sideways. (see p.11, L.252 -253, and our response to reviewer 1, specific comment #23.

17. p.8, l. 181-182: The term *second half* for the season period is confusing here, because you use the same term below to describe the half of a game. Please revise.

REPLY: This has been addressed. We have changed second half to ‘last four months’ here (see p.12, L.269)

18. p.9, l. 188: what do you understand by *extensive knowledge*? Please clarify.

 REPLY: We now explain that the coding scheme was developed in consultation with the club’s head of performance and data analyst. They each have > 7 years of experience with soccer event data (see p.10, L.212-226). 

19. p.9, l. 192: replace or define *events*.

 REPLY: We have replaced events with ‘performance indicators’ here (see p.10, 221, and our reply to reviewer 1, specific comment #29)

20. p.11, l. 213: change the reference and the information in brackets.

REPLY: this has been addressed (see p.14, L. 291)

Results

21 p.12, l. 238-240: I think this belongs to the material and method part, more specifically to participants.

 REPLY: this has been addressed, we have moved this to the statistical analysis section (see p.16, L.333-339).

22. p.12, l. 245-246: Could you provide a range of your correlations?

REPLY: we have added confidence intervals here (see p.17, L.372 & 374)

23. p.13, l. 253-255: The fact that you have removed *aerial duels” before you can just present the final correlation

 REPLY: While we agree that we should avoid redundancies, we believe that the relatively low number of events on aerial duels in the SSGs is an important component of the discussion on the degree representatives of the SGGs. Therefore, we still discuss the correlation with aerial duels (see P. 17, L. 369 - 371, as well as include aerial duels in Fig 1. 

24. p.13., l. 261-271: Think about to place this section in the methods part (e.g. the new “statistical analyses” section)

 REPLY: We have moved this part to the statistical analysis section in the revised manuscript (see p.16, from L. 329)

25. p.15, l. 294-297: Please rephrase the part with the sprint distances to clarify the results

REPLY: We have rephrased this sentence (see p.20, L.402 – 404)

Discussion

26. p.17., l. 339-351: Please think about the effect that the same experts evaluate the same players both in SSGs and 11 vs.11

 REPLY: Agreed, we have reflected on this fact in the limitations section of the revised manuscript (see p. 24, L.504-509) 

27. p.18, l. 352-354: see recommendation in general comments

 REPLY: see our reply to your general comment #6

Tables and Figures

Table 1:

28. I prefer to place the column *Offensive/Defensive aerial duel* at the end because of your exclusion of these indicators

 REPLY: this has been addressed (see p.13, Table 2)

Table 2:

29. Please adapt the statistical *n* for all indicators

 REPLY: this has been addressed (see p. 19-20, Table 4 and 5)

Figure 1:

30. I would remove the aerial duels in this figure

REPLY: see our reply to your specific comment #23

References not in manuscript

Larkin, P., & O’Connor, D. (2017). Talent identification and recruitment in youth soccer: Recruiter’s perceptions of the key attributes for player recruitment. PLoS ONE, 12(4), e0175716. https://doi.org/10.1371/journal.pone.0175716

Roberts, S. J., McRobert, A. P., Lewis, C. J., & Reeves, M. J. (2019). Establishing consensus of position-specific predictors for elite youth soccer in England. Science and Medicine in Football, 3(3), 205–213. https://doi.org/10.1080/24733938.2019.1581369

---

## [Decision Letter · Decision Letter 1]

24 Jul 2020

PONE-D-20-04681R1

The validity of small-sided games in predicting 11-vs-11 soccer game performance

PLOS ONE

Dear Dr. Bergkamp,

Thank you for submitting your manuscript to PLOS ONE. After careful consideration, we feel that it has merit but does not fully meet PLOS ONE’s publication criteria as it currently stands. Therefore, we invite you to submit a revised version of the manuscript that addresses the points raised during the review process.

Please make the minor changes outlined by the reviewers.

We look forward to receiving your revised manuscript.

Kind regards,

Caroline Sunderland

Academic Editor

PLOS ONE

Reviewers' comments:

Reviewer's Responses to Questions

**Comments to the Author**

1. If the authors have adequately addressed your comments raised in a previous round of review and you feel that this manuscript is now acceptable for publication, you may indicate that here to bypass the “Comments to the Author” section, enter your conflict of interest statement in the “Confidential to Editor” section, and submit your "Accept" recommendation.

Reviewer #1: (No Response)

Reviewer #2: All comments have been addressed

2. Is the manuscript technically sound, and do the data support the conclusions?

Reviewer #1: Yes

Reviewer #2: Yes

3. Has the statistical analysis been performed appropriately and rigorously? 

Reviewer #1: Yes

Reviewer #2: Yes

4. Have the authors made all data underlying the findings in their manuscript fully available?

Reviewer #1: Yes

Reviewer #2: Yes

5. Is the manuscript presented in an intelligible fashion and written in standard English?

Reviewer #1: Yes

Reviewer #2: Yes

6. Review Comments to the Author

Reviewer #1: Review for the revised manuscript „The validity of small-sided games in predicting 11-vs-11 soccer game performance“(PONE-D-20-04681R1)

General comments:

Thank you for submitting your revised manuscript to PLoS One. The study still presents an issue that is relevant to the field of talent research in soccer. From my point of view, the authors provide an extensive and accomplished revision of their manuscript for which they should be complimented. They addressed most of the reviewers’ comments in an adequate and satisfying way. Consequently, I would recommend the paper to be published, after some further smaller concerns have been addressed.

Specific comments:

1) Abstract, l23.ff: Please revise the first sentence as “performance” is too general and “in soccer talent” seems a bit confusing. Suggestion: Predicting game performance has been a major focus within talent identification and development/talent promotion”

2) Abstract, l.33: specify predictive validity here: “predictive validity for 11-vs-11 game performance”.

3) Abstract, l.41: remove “ability”

4) L.50 and L. 437: remove hyphen before “identification”/” development”

5) L.69: provide an example for these inconsistencies regarding performance levels.

6) L.81-84: provide reference

7) L.134: change the order to “concurrent or predictive” in line with the enumeration of the cited studies.

8) L.138: present r’s in italics.

9) L.164: Think about removing “the oldest” as this could cause confusion. Insert “… from under-15 years (U15), U17, U19, and U23 teams of a …” instead.

10) L.172: replace “away from” the average by “less than”?

11) L.174: remove “under-15 years here” (see comment 9).

12) Table 1: Title: “in brackets” instead of “between parentheses”?

13) Table 1: please provide sample sizes here as well.

14) L.270: Although it seems to be clear, please refer to the part “notational analyses” or write directly that you assessed the criterion also by performing notational analysis.

15) L.326: remove “on”?

16) L.343 ff.: provide a reference.

17) L.372: I wonder why you did not perform an additional Chi-square GoF analysis as well (after removing aerial duels. This would be consistent and would strengthen the thread of this part.

18) L.477ff/L.490ff: I am not finally convinced with the reaction on the comments of R1 and R2 regarding the reasons why correlations of the isolated physiological and motor tests with 11vs11 game performance were quite small. I am convinced that these small values are mainly since 11vs11 performance was assessed solely by on-ball-parameters. An integration of physiological and tactical (off-ball) would likely would have led to higher correlations. I appreciate, that the authors already elaborated on this, but would prefer that this fact is more highlighted and discussed. In my point of view, this could happen within the paragraph L.477ff or within L.490ff and would further strengthen the manuscript.

19) L.504 ff: Think about including a short part here that discusses your way of assessing the 11vs11 performance indicators (i.e., notational analyses). This proceeding is in parts a subjective one. Maybe other techniques (as mentioned in the introduction) could provide additional objective information that could improve the assessment of the criterion. This could be a perspective for future researchers.

Reviewer #2: 

It seems to me that the authors invested a lot of time and thoughts to revise the manuscript extensively based on the reviewers’ comments. Therefore, I would recommend the paper for publishing in PLOS ONE. There are only a few comments that should be considered before publishing the manuscript.

SPECIFIC COMMENTS

• p.3., l. 54 and ff.: use *sprint* instead of *speed*

• p.4., l. 103: insert spaces between > and 20m

• p.4., l. 113: what is about *motor skills*?

• p.5., l. 125: use the same row of the enumeration such as on p. 4., l. 113.

• p.5., l. 133: could you provide age groups investigated in this study by Olthof et al.? such as on p. 4., l. 113.

• p.8., Table 1: Thank you for providing this table. Please indicate the number of players per age group.

• p.15., l. 326: delete *on*

• p.16., l. 334: name the specific test in brackets.

---

## [Author Response · Author response to Decision Letter 1]

3 Sep 2020

Reviewer #1

General comments:

Thank you for submitting your revised manuscript to PLoS One. The study still presents an issue that is relevant to the field of talent research in soccer. From my point of view, the authors provide an extensive and accomplished revision of their manuscript for which they should be complimented. They addressed most of the reviewers’ comments in an adequate and satisfying way. Consequently, I would recommend the paper to be published, after some further smaller concerns have been addressed.

REPLY: Thank you for your positive evaluation of the manuscript. We are glad to hear that you still consider the manuscript relevant, and we believe that the quality of the manuscript has further improved. Please see our reply to your suggestions below.

Specific comments:

1) Abstract, l23.ff: Please revise the first sentence as “performance” is too general and “in soccer talent” seems a bit confusing. Suggestion: Predicting game performance has been a major focus within talent identification and development/talent promotion”

REPLY: Agreed, we have changed this sentence to “Predicting performance in soccer games has been a major focus within talent identification and development” (see p.2 L.24-25)

2) Abstract, l.33: specify predictive validity here: “predictive validity for 11-vs-11 game performance”.

 REPLY: This has been addressed (see p.2, L.33-34)

3) Abstract, l.41: remove “ability”

 REPLY: This has been addressed (see p.2, L.42)

4) L.50 and L. 437: remove hyphen before “identification”/” development”

REPLY: This has been addressed (see p.3 L.50 and p.22, L.443)

5) L.69: provide an example for these inconsistencies regarding performance levels.

 REPLY: We have added the following example from the study by Swann et al. (2016): “For example, definitions of elite athletes have ranged from international to regional level competitors, and strongly depend on the competitiveness of the sport in the athlete’s country” (see p.3-4, L.69-71)

6) L.81-84: provide reference

 REPLY: this has been addressed (see p. 4, L.86)

7) L.134: change the order to “concurrent or predictive” in line with the enumeration of the cited studies.

 REPLY: this has been addressed (see p.6, L.138)

8) L.138: present r’s in italics.

REPLY: this has been addressed (see p.6, L.142)

9) L.164: Think about removing “the oldest” as this could cause confusion. Insert “… from under-15 years (U15), U17, U19, and U23 teams of a …” instead.

 REPLY: this has been addressed. Since we already specify in L.135 that U13 stands for under-13 year old players, we have shortened it to U15 here (see P.8, L.169-170)

10) L.172: replace “away from” the average by “less than”?

REPLY: this has been addressed; we have changed ‘away from’ to ‘below’ (see p.8, L.176)

11) L.174: remove “under-15 years here” (see comment 9).

REPLY: this has been addressed (see p.8, L.178)

12) Table 1: Title: “in brackets” instead of “between parentheses”?

REPLY: this has been addressed (see p.8, Table 1)

13) Table 1: please provide sample sizes here as well.

REPLY: this has been addressed (see p.8, Table 1)

14) L.270: Although it seems to be clear, please refer to the part “notational analyses” or write directly that you assessed the criterion also by performing notational analysis.

REPLY: this has been addressed; we have emphasized that we used the same notational analysis procedure for analyzing the 11-vs-11 games as for the SSGs (see p.12, L.273-274)

15) L.326: remove “on”?

REPLY: this has been addressed (see p.16, L.331)

16) L.343 ff.: provide a reference.

REPLY: We have added the following reference here: Borenstein et al. (2010). A basic introduction to fixed-effect and random-effects models for meta-analysis. Research Synthesis Methods, 1(2), 97–111. (see p.17, L.350)

17) L.372: I wonder why you did not perform an additional Chi-square GoF analysis as well (after removing aerial duels. This would be consistent and would strengthen the thread of this part. 

REPLY: Agreed, we have performed an additional Chi-square goodness of fit test, and report its results in the revised manuscript (see p.18, L.378-379) 

18) L.477ff/L.490ff: I am not finally convinced with the reaction on the comments of R1 and R2 regarding the reasons why correlations of the isolated physiological and motor tests with 11vs11 game performance were quite small. I am convinced that these small values are mainly since 11vs11 performance was assessed solely by on-ball-parameters. An integration of physiological and tactical (off-ball) would likely would have led to higher correlations. I appreciate, that the authors already elaborated on this, but would prefer that this fact is more highlighted and discussed. In my point of view, this could happen within the paragraph L.477ff or within L.490ff and would further strengthen the manuscript.

 REPLY: We have highlighted this issue more clearly in the revised manuscript, and added that: “Integrating such (physiological) measures into our on-ball 11-vs-11 performance metrics could have increased the predictive validities of the physiological and motor tests [59].” (see p.25, L.508 - 510). 

19) L.504 ff: Think about including a short part here that discusses your way of assessing the 11vs11 performance indicators (i.e., notational analyses). This proceeding is in parts a subjective one. Maybe other techniques (as mentioned in the introduction) could provide additional objective information that could improve the assessment of the criterion. This could be a perspective for future researchers.

 REPLY: Agreed, we elaborate on the limitations of notational analysis in the revised manuscript: “Other limitations pertain to the notational analysis method used to assess soccer performance. This is a relatively intensive method to assess performance and its reliability depends on a common interpretation of indicators by each coder. Although the reliability was acceptable in our study, it is almost unavoidable that particular definitions of indicators (e.g., ‘applying pressure’) leave room for interpretation. Additionally, using the same observers to code both the predictor and criterion data could have positively affected the correlations between the indicators. Integrating physiological or tactical information derived through local or global positioning systems into the predictor or criterion may offer more reliable information. This could improve soccer performance assessments, and future research should consider if this is feasible.” (see p.25 – 26, L.512 – 521)

Reviewer #2: 

It seems to me that the authors invested a lot of time and thoughts to revise the manuscript extensively based on the reviewers’ comments. Therefore, I would recommend the paper for publishing in PLOS ONE. There are only a few comments that should be considered before publishing the manuscript.

REPLY: Thank you for your positive evaluation of the manuscript. We are glad to hear that you would recommend the paper for publication. Please see our reply to your comments below. 

SPECIFIC COMMENTS

• p.3., l. 54 and ff.: use *sprint* instead of *speed*

REPLY: Given that we would like to refer to the attribute here, rather than solely the test, we have changed ‘speed’ to ‘sprint speed’ in the revised manuscript (see p.3, L.54; p.5, L.103 & L.105)

• p.4., l. 103: insert spaces between > and 20m

REPLY: this has been addressed (see p.5, L.103 & L.104)

• p.4., l. 113: what is about *motor skills*?

REPLY: this has been addressed; we have added motor skills here (see p.5, L. 114)

• p.5., l. 125: use the same row of the enumeration such as on p. 4., l. 113.

REPLY: this has been addressed (see p.6, L.127 -128)

• p.5., l. 133: could you provide age groups investigated in this study by Olthof et al.? such as on p. 4., l. 113.

REPLY: this has been addressed; we have added the age groups here (see p.6, L.135 – 136)

• p.8., Table 1: Thank you for providing this table. Please indicate the number of players per age group.

REPLY: this has been addressed (see p.8 Table 1)

• p.15., l. 326: delete *on*

REPLY: this has been addressed (see p.16, L.331)

• p.16., l. 334: name the specific test in brackets.

REPLY: this has been addressed (see p. 17, L.339)

---

## [Editor Report · Decision Letter 2]

7 Sep 2020

The validity of small-sided games in predicting 11-vs-11 soccer game performance

PONE-D-20-04681R2

Dear Dr. Bergkamp,

We’re pleased to inform you that your manuscript has been judged scientifically suitable for publication and will be formally accepted for publication once it meets all outstanding technical requirements.

Kind regards,

Caroline Sunderland

Academic Editor

PLOS ONE
---

## [Editor Report · Acceptance letter]

11 Sep 2020

PONE-D-20-04681R2 

The validity of small-sided games in predicting 11-vs-11 soccer game performance  

Dear Dr. Bergkamp:

I'm pleased to inform you that your manuscript has been deemed suitable for publication in PLOS ONE. Congratulations! Your manuscript is now with our production department. 

Kind regards, 

on behalf of

Dr. Caroline Sunderland 

Academic Editor

PLOS ONE